# STDDN: A Physics-Guided Deep Learning Framework for Crowd Simulation

**Zijin Liu[1], Xu Geng[3], Wenshuai Xu[2], Xiang Zhao[2], Yan Xia[3], You Song[2]**
[1]School of Computer Science and Engineering, Beihang University, China
[2]School of Software, Beihang University, China
[3]Beijing Digital Native Digital City Research Center, Beijing, China
`{liuzijin, xu, by2321118, songyou}@buaa.edu.cn,`
`{gengxu, xiayan}@bdnrc.org.cn`

## ABSTRACT

Accurate crowd simulation is crucial for public safety management, emergency evacuation planning, and intelligent transportation systems. However, existing methods, which typically model crowds as a collection of independent individual trajectories, are limited in their ability to capture macroscopic physical laws. This microscopic approach often leads to error accumulation and compromises simulation stability. Furthermore, deep learning-driven methods tend to suffer from low inference efficiency and high computational overhead, making them impractical for large-scale, efficient simulations. To address these challenges, we propose the Spatio-Temporal Decoupled Differential Equation Network (STDDN), a novel framework that guides microscopic trajectory prediction with macroscopic physics. We innovatively introduce the continuity equation from fluid dynamics as a strong physical constraint. A Neural Ordinary Differential Equation (Neural ODE) is employed to model the macroscopic density evolution driven by individual movements, thereby physically regularizing the microscopic trajectory prediction model. We design a density-velocity coupled dynamic graph learning module to formulate the derivative of the density field within the Neural ODE, effectively mitigating error accumulation. We also propose a differentiable density mapping module to eliminate discontinuous gradients caused by discretization and introduce a cross-grid detection module to accurately model the impact of individual cross-grid movements on local density changes. The proposed STDDN method has demonstrated significantly superior simulation performance compared to state-of-the-art methods on long-term tasks across four real-world datasets, as well as a major reduction in inference latency.

## 1 INTRODUCTION

Crowd simulation (Karamouzas & Overmars, 2011; Feng et al., 2016; Mathew et al., 2019; Yang et al., 2020; Rasouli, 2021) has become a cornerstone technology in domains such as public safety management and intelligent transportation systems. It holds both significant research value and practical importance for real-world problems, such as crowd evacuation in densely populated areas and passenger flow organization in transportation hubs. Accurate simulation of crowd dynamics not only aids in optimizing spatial design and management strategies but also offers critical decision support for emergency responses to unforeseen events (Yang et al., 2020).

Existing mainstream approaches to crowd simulation generally fall into three categories: physics-based methods, data-driven deep learning approaches, and physics-guided deep learning methods (Rasouli, 2021; Zhang et al., 2022; Shi et al., 2023; Chen et al., 2024; Zhou et al., 2024). Physics-based methods simulate local behaviors like avoidance and following by characterizing physical interactions among individuals (Helbing & Molnar, 1995; Sarmady et al., 2010). These methods are grounded in solid theoretical foundations and offer strong physical interpretability. However, they are typically built upon simplified linear mechanics assumptions, making it difficult to capture the nonlinear and stochastic characteristics inherent in crowd behaviors. As a result, their performance drops significantly in high-density or highly interactive environments.

With the advancement of deep learning, data-driven methods (e.g., recurrent neural networks (Chen et al., 2018), graph neural networks (Xu et al., 2022), and diffusion models (Ho et al., 2020)) have achieved remarkable progress in trajectory prediction by learning motion patterns from large-scale trajectory datasets. For example, STGCNN (Mohamed et al., 2020) learns trajectory evolution from historical data, while MID (Gu et al., 2022) uses a diffusion mechanism to estimate the probability distribution of future trajectories. However, the absence of physical constraints in these methods often leads to unrealistic behaviors that violate fundamental physical laws, such as unnatural crowd congestion or collisions that ignore avoidance principles.

To bridge the gap between physical consistency and expressive modeling capacity, physics-guided deep learning approaches have emerged. These methods aim to integrate physical models into neural network architectures or training processes as inductive biases (e.g., PCS (Zhang et al., 2022), SPDiff (Chen et al., 2024)). While improving physical consistency, most focus on microscopic interactions and struggle to capture macroscopic density evolution, leading to error accumulation and suboptimal results. Additionally, SPDiff's diffusion-based paradigm suffers from slow inference and high computational cost, limiting its scalability.

Based on these insights, we propose a novel Spatio-Temporal Decoupled Differential Equation Network (STDDN) for crowd simulation. By incorporating the continuity equation from fluid dynamics as a structural constraint, STDDN guides the trajectory prediction process from a global perspective of density evolution. In contrast to existing physics-guided methods that primarily rely on individual-level modeling, STDDN treats the crowd as a continuous medium and reformulates trajectory evolution as a transport process in the density field. It uses neural networks to learn the coupling between the density and velocity fields, enabling macroscopic modeling of collective dynamics. Meanwhile, STDDN retains the strong representation capability of data-driven methods, enhancing both physical consistency and global interpretability while supporting end-to-end training. Experiments on multiple benchmark datasets demonstrate that the proposed method achieves higher prediction accuracy and greater inference efficiency in long-term simulation tasks, showing strong generalization ability and engineering applicability.

Key contributions of this paper include:

- A unified macro-micro coupled modeling framework. We incorporate the continuity equation as a differentiable physical constraint, achieving a tight, end-to-end integration of differential equations with deep neural networks for the task of crowd trajectory prediction. This design significantly enhances the physical consistency and global stability of the simulation.

- Physically interpretable design of a dynamic graph network. A cross-timestep dynamic graph neural network is constructed by using current velocity as incoming edges and future velocity as outgoing edges, enabling explicit modeling of density flux over time and improving the interpretability of the simulation process.

- Two differentiable structures for enhanced physical consistency. We propose a differentiable density mapping module based on radial basis functions and a continuous cross-grid detection module. These designs mitigate gradient discontinuities caused by the discretization process and ensure both mass conservation and smooth backpropagation.

- Superior accuracy and efficiency. Experimental results on four real-world datasets demonstrate that STDDN outperforms existing state-of-the-art methods in long-term crowd simulation accuracy, while also achieving a substantial increase in inference speed.

## 2 RELATED WORK

**Crowd simulation.** As discussed in the previous section, there are three main approaches in crowd simulation research: physics-based methods, data-driven deep learning methods, and physics-guided deep learning approaches. Physics-based methods were among the earliest frameworks studied (Dietrich & Köster, 2014; Helbing & Molnar, 1995; Pelechano et al., 2007; Corbetta & Toschi, 2023). For example, Social Force Model (SFM) (Helbing & Molnar, 1995) pioneered dynamic modeling of sparse crowds by formulating a resultant force system between pedestrians, the environment, and their goals. However, its underlying linear force assumptions exhibit clear limitations in high-density scenarios. To account for speed differences among pedestrians, a fine-grid cellular automaton model

(Sarmady et al., 2010) was proposed to enable more realistic crowd simulations. With advancements in data acquisition and modeling techniques, data-driven deep learning methods have gained wide adoption. PECNet employs a Conditional Variational Autoencoder (CVAE) (Ivanovic & Pavone, 2019) to generate multiple possible trajectories and samples a plausible endpoint based on observed data. STGCNN (Mohamed et al., 2020) leverages graph neural networks to model interactions with surrounding pedestrians at each time step. MID (Gu et al., 2022) directly models the distribution of future trajectories using a diffusion model and designs a Transformer-based architecture to capture temporal dependencies in historical data. Although these methods achieve good performance in trajectory prediction, their lack of physical constraints limits generalization under extreme or unforeseen scenarios, often resulting in physically implausible behaviors. In recent years, researchers have begun exploring physics-guided deep learning methods for crowd simulation (Kochkov et al., 2021). PCS (Zhang et al., 2022) incorporates the SFM as a prior and jointly trains two branches: one data-driven and one physics-based, through an interaction mechanism to enable information sharing. NSP (Yue et al., 2022) explicitly embeds physical parameters into the neural network to enhance interpretability. SPDiff (Chen et al., 2024) integrates the SFM with a diffusion model, building a physics-aware denoising network that has shown strong performance in crowd simulation tasks. However, since the SFM focuses solely on individual-level microscopic interactions, these methods suffer from error accumulation over time, especially in iterative simulations. Additionally, because crowd simulation often relies on autoregressive prediction combined with diffusion-based generation, inference becomes computationally intensive. To address these limitations, we propose a novel framework that integrates macroscopic physical laws, enabling simulation to be completed in a single forward pass. This significantly improves both efficiency and stability.

**Physics informed models.** In recent years, the integration of physical principles into neural networks has gained considerable attention in spatiotemporal prediction (Kochkov et al., 2021; Karniadakis et al., 2021; Fang et al., 2021; Luo et al., 2023; Chen et al., 2023; Zou & Guo, 2024). For traffic flow forecasting, STDEN (Ji et al., 2022) proposed a deep learning framework based on the continuity equation, embedding traffic flow dynamics into neural networks to bridge the gap between data-driven and physics-based models. ST-PEF (Ji et al., 2020) introduced spatiotemporal potential energy fields to simulate traffic flow, and its subsequent work, ST-PEF+ (Wang et al., 2022), further incorporated the field theory of human movement. AirPhyNet (Hettige et al., 2024) modeled particle diffusion and convection processes as differential equation networks, while Air-DualODE (Tian et al., 2024) integrated data-driven and physics-embedded methods, achieving advanced performance in air quality prediction. In trajectory prediction, neural differential equations have been widely applied to model agent dynamics. TrajODE (Liang et al., 2021) employ Neural ODEs to model continuous-time dynamics in trajectory data, Social ODE (Wen et al., 2022) utilized neural ordinary differential equations to characterize continuous trajectory evolution, NSDE (Park et al., 2024) proposed neural stochastic differential equations to enhance cross-domain generalization, and Social LODE (Ke et al., 2024) introduced latent ordinary differential equations to overcome the limitations of RNNs in long-sequence modeling. Inspired by these advances, we integrate continuity equations with crowd simulation by employing graph structures to connect historical trajectories with future predictions. This approach effectively decouples the spatial and temporal dimensions of trajectory forecasting, addressing key challenges in the field.

## 3 PRELIMINARIES

### 3.1 PROBLEM FORMULATION

We consider a group of $M$ pedestrians moving in an environment with $S$ static obstacles. At each time step $t$, the state of the crowd is denoted as $Q^t = \{p^t, v^t, a^t, d, h^t, E\}$, where $p^t$, $v^t$, and $a^t \in \mathbb{R}^{M \times 2}$ represent the positions, velocities, and accelerations of all pedestrians, respectively; $d \in \mathbb{R}^{M \times 2}$ denotes their destinations; $h^t = (p^{t-h:t}, v^{t-h:t}, a^{t-h:t})$ represents the recent motion history over a window of $h$ frames; and $E \in \mathbb{R}^{S \times 2}$ contains the positions of static obstacles in the environment. Crowd simulation is formulated as a spatiotemporal prediction problem, where the goal is to model the future evolution of individual trajectories over time and space. The model $f_\theta$ initializes from the initial state and generates the next moment's state by entering the current state, i.e.,

$$Q^{t+1} = f_\theta(Q^t, E) \tag{1}$$

which is continuously iterated until all individuals in the crowd reach their respective destinations, completing the simulation process. To ensure physical consistency, we predict acceleration $a^{t+1}$ at each time step, and update the position and velocity using the standard integration rules: $v^{t+1} = v^t + a^t \cdot \Delta t$ and $p^{t+1} = p^t + v^t \cdot \Delta t$. The Appendix A.1 provides the domains and descriptions of all symbols used in the paper.

## 3.2 CONTINUITY EQUATION

The continuity equation (Lienhard & Lienhard, 2008; Grady et al., 2010) is one of the fundamental conservation equations in fluid mechanics, describing the conservation of mass during flow. Its differential form is expressed as:

$$\frac{\partial \rho}{\partial t} + \nabla \cdot (\rho \mathbf{v}) = 0 \tag{2}$$

where $\rho$ represents the fluid's density field, denoting the mass density at spatial position $(x, y)$ and time $t$; $\mathbf{v}$ represents the fluid's velocity field at spatial position $(x, y)$ and time $t$; $\nabla \cdot (\rho \mathbf{v})$ denotes the divergence of the mass flux $\rho \mathbf{v}$. In spatiotemporal modeling, the continuity equation naturally enables a decoupling of time and space: the temporal evolution is governed by the time derivative of density, while the spatial transport is guided by the velocity field.

## 4 METHOD

### 4.1 THE MODEL FRAMEWORK

We propose the Spatio-Temporal Decoupled Differential Network (STDDN), a physically consistent framework designed to unify the modeling of the coupling between microscopic motion and macroscopic evolution. The core idea of STDDN is to leverage the continuity equation (Eq. 2) from fluid dynamics as a strong physical prior. To this end, we employ a Neural ODE to model the evolution of the macroscopic density field driven by individual movements. To compute the time derivative required for this evolution, we innovatively embed a microscopic trajectory prediction network within it, whose predictions drive the dynamic changes in the density field. This design enables macroscopic physical laws to impose physical regularization on the microscopic trajectory prediction in an end-to-end manner, thereby significantly enhancing the simulation's stability and physical consistency. The overall framework is illustrated in Figure 1.

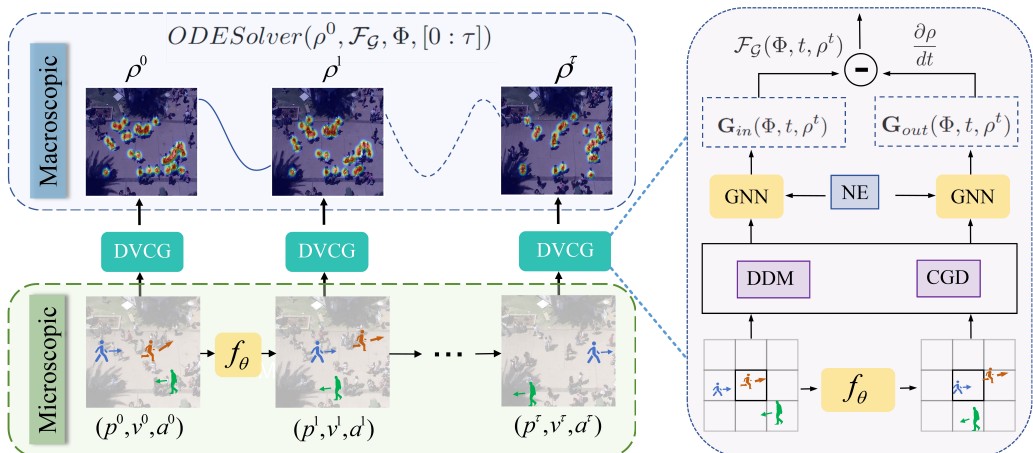

Figure 1: Overview of the STDDN framework, consisting of a microscopic trajectory prediction network (bottom left), a neural ODE-based macroscopic density evolution module (top left), and the DVCG module (right), which connects microscopic trajectories with macroscopic density and velocity fields. The DVCG module includes three components—DDM, CGD, and NE—combined in a dynamic Graph Neural Network (GNN) to model density evolution. These components build the inflow term $\mathbf{G}_{in}(\Phi, t, \rho^t)$ and outflow term $\mathbf{G}_{out}(\Phi, t, \rho^t)$, which are combined via Eq.4 to form the derivatives of the density field. Colored arrows show velocity vectors at different time steps.

We design a Density–Velocity Coupled Graph Learning (DVCG) module as the core evolution function of the neural ODE. This module takes as input the density and velocity fields at adjacent time frames, and uses a dynamic graph neural network to predict the temporal rate of change of the density field, thereby driving its continuous evolution over time. Specifically, the DVCG module consists of three key submodules: a Differentiable Density Mapping (DDM) module, a Continuous Grid Detection (CGD) module, and a Grid Node Embedding (NE) mechanism. These components jointly estimate the temporal density derivatives while maintaining physical consistency and smooth gradient flow during training.

Furthermore, STDDN enables joint training by allowing macroscopic density predictions to guide and refine the microscopic trajectory prediction module. During the inference stage, the model utilizes the trained single-step trajectory prediction network to perform autoregressive inference according to Eq. 1, generating future trajectories step by step, thus enabling high-precision simulation of crowd dynamics. It is important to note that during inference, we only require the trained model $f_\theta$ and do not rely on the ODE part.

## 4.2 DENSITY–VELOCITY COUPLED GRAPH LEARNING MODULE

Inspired by Chapman (2015), we discretize the spatial region into regular grids and use them to construct a graph structure. Leveraging the trajectory information of individual pedestrians, we propose an innovative density–velocity coupled dynamic graph learning module (DVCG), designed to efficiently compute the temporal derivative of the macroscopic density field. Specifically, the velocities of all individuals at the current time step form the set of incoming edges for the graph nodes, while the predicted velocities at the next time step constitute the outgoing edges. The flux at each grid node is determined jointly by the pedestrian velocities and the current node density. This design transforms the trajectory prediction task into a spatiotemporal density transport optimization problem, offering stronger physical interpretability. In a discrete local spatiotemporal field, the density evolution $\rho$ of a spatial grid node $i$ and its connected nodes $j$ and $k$ which are linked by the velocity vectors $\mathbf{v}_{ji}$ and $\mathbf{v}_{ik}$, at time $t$ can be expressed as:

$$\frac{\partial \rho_i}{dt} = \sum_{\{\forall j | j \to i\}} (m_t w_{ji} \|\mathbf{v}_{ji}^t\| \rho_j^t + b_{ji}) - \sum_{\{\forall k | i \to k\}} (m_{t+1} w_{ik} \|\mathbf{v}_{ik}^{t+1}(\mathbf{v}_{ik}^t; \theta)\| \rho_i^{t+1}(\mathbf{v}_{ik}^t; \theta) + b_{ik})$$

(3)

The above flux operations can be compactly represented in matrix form as:

$$\frac{\partial \rho}{dt} = \mathcal{F}_\mathcal{G}(\Phi, t, \rho^t) = \mathbf{G}_{in}(\Phi, t, \rho^t) - \mathbf{G}_{out}(\Phi, t, \rho^t)$$

$$\mathbf{G}_{in}(\Phi, t, \rho^t) = \rho^t(m_t \odot \mathbf{A}^t \odot \mathbf{W} \odot \|\mathbf{V}^t\| + \mathbf{A}^t \odot \mathbf{B})$$

(4)

$$\mathbf{G}_{out}(\Phi, t, \rho^t) = \rho^{t+1}(\mathbf{V}^t; \theta) \odot (\mathbf{1}(m_t \odot \mathbf{A}^{t+1} \odot \mathbf{W} \odot \|\mathbf{V}^{t+1}(\mathbf{V}^t; \theta)\| + \mathbf{A}^{t+1} \odot \mathbf{B})^T)$$

Here, $\mathcal{F}_\mathcal{G}$ denotes the time derivative of the density $\rho$. $\Phi$ represents all learnable parameters, including the weight matrix $\mathbf{W}$, the bias matrix $\mathbf{B}$, and $\theta$. The terms $\mathbf{v}^{t+1}(\mathbf{v}^t; \theta)$ and $\rho^{t+1}(\rho^t; \theta)$ refer to the predicted velocity and density at the next time step, respectively, as generated by a neural network $f_\theta$. The operator $\odot$ denotes element-wise (Hadamard) multiplication. The cross-grid masks $m_t$ and $m_{t+1}$ are provided by the CGD module. The matrices $\mathbf{A}^t$ and $\mathbf{A}^{t+1}$ represent the dynamic connectivity matrices at time $t$ and $t + 1$, respectively, and the terms $w_{ji}$ and $w_{ik}$ are the learnable parameters in $\mathbf{W}$, while $b_{ji}$ and $b_{ik}$ are the learnable parameters in $\mathbf{B}$. Theoretically, $f_\theta$ can be any existing microscopic prediction model. However, to ensure a fair comparison, we adopt the same network architecture for $f_\theta$ as in Chen et al. (2024). (More details about the neural network are provided in the Appendix A.2.) Since individual velocities vary over time, the constructed graph exhibits dynamic evolution characteristics. Starting from the initial density $\rho^0$, we use an ODE solver to compute the future density sequence over the $\tau$ time steps, denoted as $\rho^{1:\tau}$:

$$\rho^{1:\tau} = ODESolver(\rho^0, \mathcal{F}_\mathcal{G}, \Phi, [0 : \tau])$$

(5)

More details about the ODE solver are provided in the Appendix A.3.

## 4.3 DIFFERENTIABLE DENSITY MAPPING MODULE

The Differentiable Density Mapping (DDM) module maps continuous individual positions onto a discrete spatial grid to produce density values. Traditional hard assignment methods cause discon-

tinuous gradients at grid boundaries, hindering model optimization. To address this, we adopt a probability-based soft assignment strategy. Specifically, we first compute the coordinates of each grid center $c_i$, forming the matrix $D = [c_0, c_1, \cdots, c_{N-1}] \in \mathbb{R}^{1 \times N}$. For any predicted position $p_t$, we calculate its squared Euclidean distances to all grid centers:

$$d(p^t) = [\|p^t - c_0\|^2, \|p^t - c_1\|^2, \cdots, \|p^t - c_{N-1}\|^2].$$

These distances are then converted into a probability distribution using a temperature-controlled softmax function:

$$q_i(p^t) = \frac{\exp(-\beta \cdot |p^t - c_i|^2)}{\sum_{j=0}^{N-1} \exp(-\beta \cdot |p^t - c_j|^2)} \tag{6}$$

where $\beta$ is a temperature parameter that controls the smoothness of the distribution.

Finally, by summing the position distributions of all $K$ individuals, we obtain a continuous and differentiable density representation:

$$\rho^t = \sum_{i=1}^{K} q_i(p^t) \tag{7}$$

## 4.4 CONTINUOUS CROSS-GRID DETECTION MODULE

In crowd movement simulation, net flux arises only from trajectories crossing grid boundaries. Traditional discrete methods for statistical detection cause gradient discontinuities, hindering end-to-end training. To address this, we first obtain a continuous spatial distribution via a differentiable position-to-grid probability mapping, then quantify the extent of grid crossing by computing the Jensen-Shannon divergence between probability distributions at consecutive time steps:

$$\mathcal{J}(q(p^t)|q(p^{t+1})) = \frac{1}{2} \left[ \mathcal{KL}(q(p_t), M) + \mathcal{KL}(q(p^{t+1}), M) \right] \tag{8}$$

where $\mathcal{KL}$ denotes the Kullback-Leibler divergence, and $M = (q(p^t) + q(p^{t+1}))/2$ represents the mean distribution. The divergence values are then transformed into differentiable cross-grid masks using a temperature-controlled sigmoid activation function:

$$m = \sigma(\alpha(\mathcal{J} - \tau)) \in [0.01, 0.99] \tag{9}$$

where $\sigma(\cdot)$ is sigmoid activation, $\alpha$ is a scaling factor and $\tau$ is a threshold parameter. This formulation maintains gradient continuity while preserving physical constraints.

## 4.5 NODE EMBEDDING

The granularity of grid partitioning affects how strongly the continuity equation constrains trajectory prediction: finer grids capture more detailed cross-cell movements but incur significant memory overhead, as traditional weight matrices scale with $O(N^2)$. To address this, we propose a lightweight node embedding representation. Each grid node is associated with learnable embedding and bias vectors: $\mathbf{w} = [w_0, w_1, \cdots, w_{N-1}]^T \in \mathbb{R}^{N \times d}$, $\mathbf{b} = [b_0, b_1, \cdots, b_{N-1}]^T \in \mathbb{R}^{N \times d}$. The weight and bias matrices are then dynamically constructed via outer products: $\mathbf{W} = \mathbf{w}\mathbf{w}^T$, $\mathbf{B} = \mathbf{b}\mathbf{b}^T$. This design reduces the storage complexity to $O(N \cdot d)$, significantly improving memory efficiency while preserving modeling capacity.

## 4.6 TRAINING

By coupling the trajectory prediction network with the differential equation, the training objective jointly supervises both velocity and density through a combined loss function:

$$l_{joint} = l_{NN} + l_{ODE} = \lambda_1 \|\mathbf{v} - \mathbf{v}_\theta\| + \lambda_2 \|\rho - \rho_\theta\| \tag{10}$$

where $\lambda_1$ and $\lambda_2$ are balancing coefficients that control the relative importance of velocity prediction accuracy and density evolution consistency, respectively. The full training procedure is provided as pseudocode in the Appendix A.4.

## 5 EXPERIMENTS

### 5.1 EXPERIMENTAL SETUP

**Datasets.** We evaluate our framework on four open-source trajectory datasets: GC (Zhang et al., 2022), UCY (Lerner et al., 2007) which includes three sub-scenes (ZARA1, ZARA2, and UCY), and ETH (Pellegrini et al., 2009), which includes two sub-scenes (ETH and HOTEL). These datasets span diverse scene types, spatial scales, time durations, and pedestrian densities, providing a comprehensive benchmark for assessing the simulation capabilities of our framework. Following the approach of SPDiff (Chen et al., 2024), we select dense and long-duration segments for GC and UCY (300s and 216s respectively), focusing on regions with over 200 pedestrians per minute. For ETH and HOTEL, following the approach of STGCNN (Mohamed et al., 2020) we adopt the original training/test splits. Further preprocessing details, including coordinate transformation and temporal interpolation, are provided in the Appendix A.5.

**Baselines.** We divide the baseline methods into physics-based, data-driven, and physics-guided methods. Within the physics-based methods, we choose the widely used Social Force Model (SFM) (Helbing & Molnar, 1995) and Cellular Automaton (CA) (Sarmady et al., 2010) for comparison. Within the data-driven methods, we select three representative approaches recently published, including STGCNN (Mohamed et al., 2020) which utilizes graph convolutional neural networks to compute a spatiotemporal embedding, PECNet (Mangalam et al., 2020) which uses VAE to sample multi-modal endpoints and MID (Gu et al., 2022) which is based on the diffusion framework to model indeterminacy. For physics-guided methods, we select PCS (Zhang et al., 2022), whose backbone is graph networks, NSP (Yue et al., 2022), based on sequence prediction models combined with CVAE, SPDiff (Chen et al., 2024) which employs a diffusion based model for next time step prediction model.

**Metrics.** We employ four evaluation metrics to comprehensively assess the model's performance and inference efficiency. For trajectory accuracy, Mean Absolute Error (MAE) and Optimal Transport (OT) (Villani, 2021) distance are used to measure point-to-point error and trajectory shape similarity (Sanchez-Gonzalez et al., 2020; Gretton et al., 2012), respectively. These are widely used statistical metrics in physical process modeling. For inference efficiency, we adopt number of parameters (#Pars) and single-frame inference latency (Latency, measured in ms). Detailed definitions and implementation specifics are provided in the Appendix A.6, along with comparative results for additional metrics in the Appendix A.7.

### 5.2 OVERALL PERFORMANCE

To substantiate the effectiveness of our proposed framework, experiments were executed across four large-scale crowd simulation datasets, followed by a systematic comparison against existing crowd simulation methods. The empirical findings (presented in Table 1 and Table 2, more results can be found in the Appendix A.7) unequivocally demonstrate STDDN's substantial advantages over the second-best method, SPDiff, across all evaluation settings. Specifically: for the GC dataset, inference duration was reduced by 50%, concurrently yielding MAE and OT metric enhancements of 2.6% and 2.46%; on the UCY dataset, inference time saw a 90% reduction, with MAE and OT metrics boosting by an impressive 5.39% and 10.01%; on the ETH dataset, inference time decreased by 50%, alongside accuracy gains of 6.0% and 19.81%; and for the HOTEL dataset, inference time experienced a 75% reduction, while MAE and OT metrics consistently improved by 12.66% and 12.21%. Specifically, our observations are as follows: Firstly, in comparing pure data-driven methods with purely physical model approaches, STDDN achieves a superior balance within the network by more effectively integrating the two paradigms. It not only assimilates the objective laws inherent in physical models but also captures general pedestrian movement regularities. Secondly, while both SPDiff and MID utilize diffusion models to condition future trajectories on historical data, STDDN attains superior performance merely by constraining its model parameters via the continuity equation. Furthermore, single-frame simulation with diffusion models necessitates multiple forward passes, whereas STDDN requires only a single pass, leading to significantly reduced inference time and a considerable efficiency advantage over diffusion models. Lastly, despite the significant variations in pedestrian speeds observed in the ETH and HOTEL datasets, our method

Table 1: Overall performance comparison on GC and UCY datasets. The bold and underlined font show the best and the second best result respectively. Performance averaged over 5 runs.

| Model | GC | | | | UCY | | | |
|---|---|---|---|---|---|---|---|---|
| | MAE↓ | OT↓ | #Pars↓ | Latency↓ | MAE↓ | OT↓ | #Pars↓ | Latency↓ |
| CA | 2.7080 | 5.4990 | - | - | 8.3360 | 79.4200 | - | - |
| SFM | 1.2590 | 2.1140 | - | - | 2.5390 | 6.5710 | - | - |
| STGCNN | 8.1608 | 15.8372 | 7.56K | 8.0539 | 7.5121 | 18.7721 | 7.56K | 8.0539 |
| PECNet | 2.0669 | 4.3054 | 2.10M | 51.3781 | 3.7694 | 16.1412 | 1.23M | 49.5681 |
| MID | 8.4257 | 35.1797 | 3.52M | 245.7762 | 8.2915 | 47.8711 | 2.47M | 276.8964 |
| PCS | 1.0320 | 1.5963 | 0.97M | 30.4784 | 2.3134 | 6.2336 | 0.62M | 21.2818 |
| NSP | 0.9884 | 1.4893 | 1.93M | 60.7866 | 2.4006 | 6.3795 | 2.14M | 31.3471 |
| SPDiff | 0.9116 | 1.3925 | 0.14M | 206.9909 | 1.8760 | 4.0564 | 0.22M | 471.0496 |
| **Ours** | **0.8875** | **1.3582** | 0.17M | 86.8537 | **1.7747** | **3.6503** | 0.07M | 44.6565 |

consistently achieved better performance, thereby further validating the efficacy of the continuity equation constraint. Visualization of generated trajectories are provided in the Appendix A.9.

Table 2: Overall performance comparison on ETH and HOTEL datasets. The bold and underlined font show the best and the second best result respectively. Performance averaged over 5 runs.

| Model | ETH | | | | HOTEL | | | |
|---|---|---|---|---|---|---|---|---|
| | MAE↓ | OT↓ | #Pars↓ | Latency↓ | MAE↓ | OT↓ | #Pars↓ | Latency↓ |
| CA | 0.7211 | 2.5318 | - | - | 0.5328 | 0.3102 | - | - |
| SFM | 0.6933 | 1.1903 | - | - | 0.4977 | 0.2371 | - | - |
| STGCNN | 3.8711 | 15.3766 | 7.56K | 8.0539 | 5.6711 | 7.3851 | 7.56K | 8.0539 |
| PECNet | 0.6533 | 0.9712 | 2.10M | 48.3381 | 0.4592 | 0.2380 | 1.93M | 31.8613 |
| MID | 4.1823 | 16.8301 | 3.79M | 73.6639 | 4.3867 | 23.7882 | 2.86M | 60.6521 |
| PCS | 0.6573 | 1.0121 | 0.62M | 33.2728 | 0.4278 | 0.2068 | 0.88M | 13.5936 |
| NSP | 0.6343 | 0.9372 | 1.57M | 40.4512 | 0.3972 | 0.1984 | 1.86M | 21.3488 |
| SPDiff | 0.5527 | 0.8706 | 0.18M | 81.4144 | 0.3380 | 0.1646 | 0.16M | 68.5705 |
| **Ours** | **0.5185** | **0.6918** | 0.20M | 30.5723 | **0.2952** | **0.1445** | 0.05M | 17.4986 |

## 5.3 ACCUMULATED ERROR ANALYSIS OF THE SIMULATION

To further investigate the error accumulation behavior of our method in long-term prediction, we analyze the temporal evolution of MAE and OT metrics on the GC and UCY datasets, and compare the results with two crowd simulation baselines: SPDiff and PCS. As shown in Figure 2, both our method and SPDiff exhibit a "rise-then-fall" trend over time. However, our method consistently achieves the lowest overall error, indicating that it is the least affected by error accumulation during long-term prediction. This performance advantage is primarily attributed to the incorporation of macroscopic physical constraints into our model, which guide it toward globally optimal solutions and effectively suppress error propagation over time. To more comprehensively evaluate our model's long-term density prediction capability, we provide additional analysis of time-accumulated density prediction errors in the supplementary materials.accumulation of density prediction errors in the Appendix A.10.

## 5.4 ABLATION STUDY

To further investigate the contributions of key components in STDDN, we conduct an ablation study on the GC and UCY datasets. Specifically, we evaluate the following variants: w/o ODE: The neural ODE constraint is removed, and the model is trained solely via a purely autoregressive scheme for trajectory prediction. w/o Cross-net: The Continuous Grid-Crossing Detection (CGD) module is removed. Since our approach is flux-based, the absence of this module significantly weakens

the physical enforcement of the continuity equation. w/o NN loss: The trajectory autoregressive training objective is discarded; the model is optimized only using the density-field-based neural ODE loss. w/o NE: Learnable node embeddings (Node Embedding) and bias terms are removed. Trans: The dynamically constructed adjacency structure—derived from predicted trajectories—is replaced with a static attention-based learned weight matrix. Dopri5 / RK4: The default Euler solver is replaced with higher-order ODE solvers, namely Dormand–Prince 5 (Dopri5) and Runge–Kutta 4 (RK4), respectively. Discrete NN: The neural ODE solver is replaced by an explicit first-order discrete update implemented via a residual connection: $\rho^{t+1} = \rho^t + \Delta t \cdot \mathcal{F}_\mathcal{G}(\Phi, t, \rho^t)$, which corresponds to the forward Euler discretization of the continuity equation: $\frac{\partial \rho}{\partial t} = \mathcal{F}_\mathcal{G}(\Phi, t, \rho^t)$.

The experimental results are presented in Table 3, leading to the following conclusions: First, the significant performance drop in w/o ODE demonstrates that incorporating the continuity equation as a physical constraint effectively mitigates the accumulation of errors inherent in purely autoregressive models during long-horizon crowd simulation, thereby validating the necessity of the differential equation framework. Second, the substantial degradation observed in w/o Cross-net indicates that accurately modeling the flux generated as agents cross grid boundaries is cru-

Table 3: Ablation study on different parts of model design.

|  | GC | | UCY | |
| --- | --- | --- | --- | --- |
|  | MAE | OT | MAE | OT |
| w/o ODE | 1.3784 | 2.4956 | 2.4867 | 6.0586 |
| w/o Cross-net | 0.9784 | 1.4732 | 1.8926 | 4.9532 |
| w/o NN loss | 1.2387 | 2.3466 | 1.9327 | 4.2514 |
| w/o NE | 0.8921 | 1.3881 | 1.7917 | 3.7131 |
| Trans | 0.8901 | 1.3611 | 1.7833 | 3.7055 |
| Dopri5 | 1.1315 | 1.7422 | 1.9654 | 5.2318 |
| RK4 | 1.2311 | 1.8625 | 2.0381 | 5.4581 |
| Discrete NN | 0.8875 | 1.3582 | 1.7747 | 3.6503 |
| **Ours** | **0.8875** | **1.3582** | **1.7747** | **3.6503** |

cial for enforcing mass conservation, further confirming the effectiveness of the continuity constraint. Third, the results of w/o NN loss reveal that a model driven solely by physical laws struggles to capture the complex spatiotemporal dynamics of real pedestrian motion, highlighting the synergistic advantage of combining data-driven learning with physical priors. Moreover, comparisons between w/o NE and Trans show that the trajectory-driven dynamic graph structure, together with learnable node embeddings, more effectively encodes spatial relationships and interaction semantics than static attention mechanisms. Notably, although Dopri5 and RK4 offer higher theoretical numerical accuracy, they lead to performance degradation in our task. This is because higher-order solvers introduce intermediate interpolated states that do not align with the actual observation time steps, increasing computational overhead and raising the risk of overfitting. In contrast, the Euler solver naturally aligns with our autoregressive-style discrete-time modeling, achieving an optimal trade-off between computational efficiency and simulation accuracy. Finally, Discrete NN achieves performance comparable to our full method, indicating that our approach fundamentally models inter-frame crowd flow transitions on a discrete spatiotemporal grid, rather than a smooth, continuous dynamical process.

## 5.5 SENSITIVITY STUDY

We conducted a sensitivity analysis on the key hyperparameters of our framework, including grid size, ODE time steps $\tau$, loss balancing coefficients $\lambda_1$ and $\lambda_2$, and node embedding dimensions. The MAE results of GC dataset are shown in Figure 3 (More results of sensitivity analysis on different datasets can be found in the Appendix A.11.). The results show that variations in grid size, ODE time steps, and embedding dimensions all have a certain impact on the final prediction performance, which further demonstrates the effectiveness of the continuity equation as a physical constraint within the model. Additionally, while different values of the loss balancing coefficients lead to varying results, setting the two terms with comparable weights enables a better trade-off between data-driven learning and physical modeling, thereby achieving improved predictive accuracy and highlighting the value of incorporating physical constraints into autoregressive prediction models.

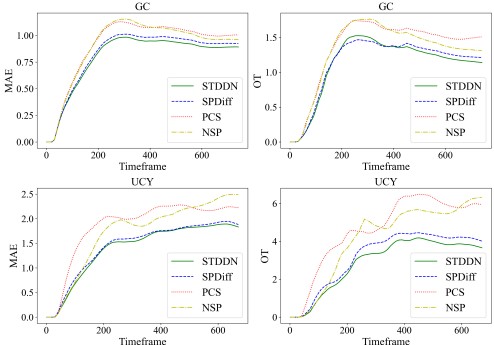

Figure 2: Accumulate error of the simulation, using MAE and OT as metrics.

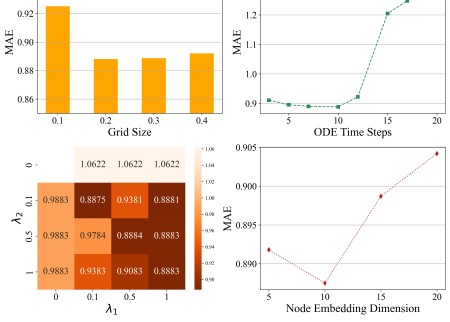

Figure 3: Sensitivity analysis on the GC dataset for grid size, ODE time steps $\tau$, loss balancing coefficients ($\lambda_1$ and $\lambda_2$), and node embedding dimension, using MAE as the evaluation metric.

## 6 CONCLUSION

This paper proposes STDDN, a novel crowd simulation framework that integrates macroscopic physical constraints with deep learning. By leveraging the continuity equation, STDDN systematically models the coupled evolution of crowd density and velocity fields, effectively mitigating error accumulation and instability commonly seen in microscopic approaches. Through the design of differentiable density mapping module, continuous cross-grid detection module, and grid node embedding mechanism, STDDN enables high-precision, end-to-end simulation. Experimental results demonstrate that our method significantly outperforms existing mainstream methods on long-term trajectory simulation tasks across multiple real-world datasets, achieving not only superior performance but also a substantial reduction in inference time. This method offers a physics-consistent and interpretable deep learning paradigm for crowd modeling, showing strong potential for practical applications.

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

# A    APPENDIX

## A.1    NOTATIONS

An overview of the notations used in this paper, along with their domains and descriptions, is provided in Table 4.

Table 4: List of major symbols and descriptions.

| Sym. | Domain | Descriptions |
|------|--------|--------------|
| $K$ | $\mathbb{R}$ | Number of pedestrian |
| $N$ | $\mathbb{R}$ | Number of nodes |
| $\rho^t$ | $\mathbb{R}^N$ | Density of nodes at time $t$ |
| $m_t$ | $\mathbb{R}^k$ | cross-grid masks of K pedestrianat time $t$ |
| $\mathbf{A}$ | $\{0,1\}^{N \times N}$ | Dynamic adjacency matrix |
| $\mathbf{W}$ | $\mathbb{R}^{N \times N}$ | Learnable weight matrix |
| $\mathbf{B}$ | $\mathbb{R}^{N \times N}$ | Learnable bias matrix |
| $\mathbf{v}$ | $\mathbb{R}^{N \times N \times 2}$ | Velocity tensor matrix |
| $\|\mathbf{v}\|$ | $\mathbb{R}^{N \times N}$ | Velocity magnitude matrix |
| $\mathbf{G_{in}}$ | $\mathbb{R}^{1 \times N}$ | The inflow vector of all nodes |
| $\mathbf{G_{out}}$ | $\mathbb{R}^{1 \times N}$ | The outflow vector of all nodes |
| $\mathbf{1}$ | $\{1\}^{1 \times N}$ | All-ones row vector |

## A.2    NEXT TIMEFRAME TRAJECTORY PREDICTION MODEL

To ensure the fairness of experimental comparisons, we adopt the same denoising network architecture as used in the sub-optimal baseline SPDiff, with all diffusion step embeddings fixed to 0. The overall network architecture, as illustrated in the Figure 4, consists of three key representation learning components: encoding of historical trajectories, modeling of interactions with surrounding pedestrians, and attraction modeling toward the destination.

The network employs the Equivariant Graph Convolution Layer (EGCL) module for message passing. Its computation logic is as follows: At the $l$ layer, the node embedding $h^l$, position embedding $p^l$, velocity embedding $v^l$, and acceleration embedding $a^l$ are used as inputs to compute the updated embeddings $h^{l+1}$, $p^{l+1}$, $v^{l+1}$, and $a^{l+1}$. The update process consists of the following steps:

$$m_{ij} = \phi_e(h_j^l, h_j^l, \|p_i^l - p_j^l\|^2), \tag{11}$$

$$a_i^{l+1} = \phi_a(h_j^l)y_{k,i} + \sum_{j \in N(i)} \frac{1}{d_{ij}}(p_i^l - p_j^l)\phi_p(m_{ij}), \tag{12}$$

$$v_i^{l+1} = v_i^l + a_i^{l+1}, p_i^{l+1} = p_i^l + v_i^{l+1}, \tag{13}$$

$$m_i = \sum_{j \in N(i)} m_{ij}, h_i^{l+1} = \phi_h(h_l^l, m_i) \tag{14}$$

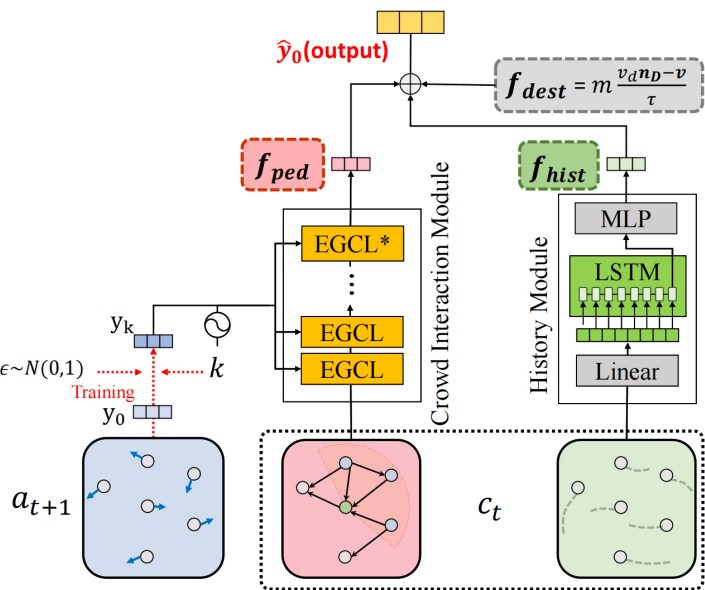

Figure 4: The detailed of next trajectory prediction model.

### A.3 ODESOLVER

In our experiments, we adopt the Euler method as the numerical solver for ODEs. The solver is configured with relative tolerance (rtol) of 1e-4 and absolute tolerance (atol) of 1e-3. The crowd trajectory and density data are sampled from fixed-frame-rate videos, resulting in discretized time steps with consistent intervals. The single-step update mechanism of the Euler method allows precise alignment with each frame's timestamp, eliminating the need for multiple sub-steps within a single time interval and avoiding inconsistencies with the actual temporal resolution. Since the Euler method requires only one forward computation per frame, it offers low computational complexity and minimal memory overhead, making it well-suited for long-duration, multi-scene, and large-scale crowd simulation tasks. Moreover, neural ODEs are easily implemented using the torchdiffeq package.

### A.4 TRAINING ALGORITHM PSEUDOCODE

We provide the pseudocode for the training algorithm, as shown in Algorithm 1.

---

**Algorithm 1** Train

---

1: Trajectory prediction model $f_\theta$
2: **while** not converged **do**
3:     Draw $(p^{0:\tau}, v^{0:\tau}, a^{0:\tau})$ from dataset
4:     Compute initial crowd density $\rho^0$ by Eq.(6) and Eq.(7)
5:     **for** $t \in [0:\tau]$ **do**
6:         Predict next trajectory $p^{t+1}$ by $f_\theta$
7:         Execute continuous cross-grid detection module by Eq.(8) and Eq.(9)
8:         Compute density net flux by Eq.(4)
9:         Execute the single-step ODE solver in Eq.(5)
10:     **end for**
11:     Compute joint loss $l_{joint}$ by Eq.(10)
12: **end while**

---

Table 5: The basic statistics of the datasets.

| Statistics | GC | UCY | ETH | HOTEL | ZARA1 | ZARA2 |
|---|---|---|---|---|---|---|
| Average duration of a trajectory$(s)$ | 11.02 | 13.25 | 6.10 | 6.72 | 13.92 | 19.06 |
| Average pedestrian per minute $(min^{-1})$ | 203 | 132 | 49 | 35 | 44 | 36 |
| Pedestrian density $(m^{-2})$ | 0.094 | 0.058 | 0.016 | 0.012 | 0.149 | 0.241 |
| Average speed $(m \cdot s^{-1})$ | 1.155 | 1.072 | 2.293 | 1.038 | 1.071 | 0.793 |
| Std of the average speed $(m \cdot s^{-1})$ | 0.565 | 0.646 | 0.807 | 0.695 | 0.134 | 0.349 |
| simulate time steps | 725 | 651 | 5776 | 9006 | 4481 | 5231 |

## A.5 DATASET DETAILS

To evaluate the effectiveness of our proposed model, we conduct experiments on six real-world pedestrian trajectory datasets constructed from surveillance video recordings. These datasets differ in scene complexity, crowd density, and temporal resolution. The key statistics are summarized in Table 5. Detailed descriptions are as follows:

**GC.** The GC dataset contains 12680 annotated trajectories within an image coordinate system covering approximately 30×35 meters. We select a 300-second subset within a 20×20 meter area that includes rich pedestrian interactions. The original sampling rate is 1.25 Hz ($\Delta t = 0.8s$). To enhance temporal resolution and reduce interpolation error, we apply cubic interpolation using SciPy, resulting in a new timestep of 0.08 s. The coordinates are converted from image space to world coordinates using a provided mapping.

**UCY.** The original UCY dataset includes three sub-scenes: ZARA1, ZARA2, and UCY. The ZARA1 scene contains 148 pedestrians over 360 seconds and covers a 16×12 meter area. The ZARA2 scene contains 204 pedestrians over 420 seconds and covers a 16×12 meter area. The UCY scene contains 528 pedestrians over 216 seconds and covers a 23×25 meter area. Trajectories in three scenes are converted to world coordinates and interpolated three times to reduce the timestep from 0.4 s to 0.08 s, improving temporal continuity and simulation accuracy.

**ETH.** The ETH dataset contains two sub-scenes: ETH and HOTEL. The ETH scene spans 464 seconds with 360 pedestrians in an 80×20 meter space, while the HOTEL scene spans 692 seconds with 389 pedestrians over 50×30 meters. We follow the original train/test splits provided in prior work. Both scenes undergo coordinate transformation and cubic interpolation to unify the timestep to 0.08 s, consistent with the GC and UCY datasets.

We divide each dataset into training and testing sets based on time. For the GC dataset, the training-to-testing ratio is set to 4:1, while for the UCY, ZARA1 and ZARA2 datasets, it is 3:1. For ETH and HOTEL, the original training and validation sets are merged into a single training set, and evaluation is performed on the original test set.

## A.6 METRICS

**MAE**: The MAE measures the average magnitude of errors between predicted and true positions, and is defined as:

$$MAE = \frac{1}{N} \sum_{i=1}^{N} \|p - \hat{p}\|_2 \tag{15}$$

where $N$ is the total number of prediction instances, $p$ denotes the predicted position, $\hat{p}$ is the corresponding ground-truth position, and $\|\cdot\|$ denotes the matrix norm.

**OT**: It quantifies the transportation cost between two distributions $P$ and $Q$, which can be interpreted as the distance between the predicted and ground-truth trajectory distributions:

$$OT(P\|Q) = \inf_{\pi} \int_{X \times Y} \pi(x,y)c(x,y)dxdy,$$
$$s.t. \int_{Y} \pi(x,y)dy = P(x), \int_{X} \pi(x,y)dx = Q(y) \tag{16}$$

where $\pi(x, y)$ is the transportation plan approximated from $P(x)$ and $Q(y)$ using the Sinkhorn algorithm, and $c(x, y) = \|x - y\|_2^2$. The domain $X = Y$ corresponds to the simulation duration, while $P$ and $Q$ denote the predicted and ground-truth trajectory distributions.

**MMD**: It measures the difference between two distributions by comparing the mean embeddings in a reproducing kernel Hilbert space (RKHS). It is often used to assess the similarity between predicted and real trajectory distributions:

$$MMD^2(P\|Q) = \|\mathbb{E}_{x \sim P}[\phi(x)] - \mathbb{E}_{y \sim Q}[\phi(y)]\|^2 \tag{17}$$

where $P$ and $Q$ represent the predicted and ground-truth trajectory distributions, and $\phi(\cdot)$ denotes the mapping to the high-dimensional feature space induced by a kernel. In this work, we use the Gaussian kernel to compute the empirical MMD.

**DTW**: It measures the similarity between two trajectory sequences by aligning them non-linearly along the time axis, allowing for time warping. It is defined as:

$$DTW(X, Y) = \min_{w \in \mathcal{W}} \sum_{(i,j) \in w} \|x_i - y_j\| \tag{18}$$

where $X = (x_1, ..., x_n)$ and $Y = (y_1, ..., y_m)$ are two trajectory sequences, $\mathcal{W}$ is the set of all possible alignment paths, and $\|\cdot\|$ denotes the Euclidean distance. DTW enables alignment of trajectories with varying speeds, thus better capturing their shape similarity.

**FDE**: It measures the distance between the predicted final destination and the ground-truth final destination at the end of the prediction horizon. It is defined as:

$$FDE = \frac{1}{N} \sum_{n \in N} \|\hat{p}_t^n - p_t^n\|_2, t = T_p \tag{19}$$

where $\hat{p}_t^n$ is the predicted final position at the last prediction timestep $T_p$, and $p_t^n$ is the corresponding ground-truth final position.

**#Colli**: We count two pedestrians with a distance less than 0.5m as a collision and take the summation of collisions in all frames as Collision. But considering that pedestrians could walk with their friends, to whom they won't keep a large social distance, We take the pair of pedestrians that have a collision in more than 2 seconds as friends and do not count the collisions between them into Collision.

**DEA**: To assess whether the model generates unrealistically dense local clusters, we define the local density of pedestrian $i$ at time $t$ as:

$$d_i^t = \sum_{j=1, j \neq i}^{N} \mathbb{I}(\|p_i^t - p_j^t\|_2 \leq R) \tag{20}$$

,i.e., the number of neighbors within radius $R = 1$ meter. Using the mean local density from ground-truth trajectories, $\mu_{\text{GT}}$, as a threshold, we compute:

$$DEA = \frac{1}{N \cdot T} \sum_{i=1}^{N} \sum_{t=1}^{T} \mathbb{I}(\hat{d}_i^t > \mu_{\text{GT}}) \tag{21}$$

**#Pars**: This metric measures the total number of learnable parameters (weights and biases) in the model, directly reflecting the model's static size and storage complexity. A lower parameter count indicates a more lightweight model, making it easier to deploy. The unit is typically in thousands (K) or millions (M).

**Latency**: This metric measures the actual time required for the model to complete a single-frame prediction on a specific hardware platform. It is a hardware-dependent metric that directly reflects the real running speed of the model. All latency tests in this paper were conducted on an NVIDIA RTX 4090, with the unit in milliseconds (ms).

**GFLOPs**: This metric measures the theoretical total number of floating-point operations required for the model to generate the next frame from its current state. It is a hardware-independent metric that fairly reflects the complexity of the model's algorithm itself.

**FPS**: This metric measures the number of simulation frames (Frames Per Second) that the model can generate per second in continuous autoregressive mode. It directly evaluates the model's throughput and performance in real-world applications, serving as the gold standard for assessing whether the simulation system can meet real-time requirements.

## A.7 Additional Experimental Results

To gain deeper insights into the strengths of our method, we extend the evaluation beyond standard metrics and analyze its performance in terms of trajectory shape fidelity, inference efficiency, and physical plausibility across six benchmark datasets: GC, UCY, ETH, HOTEL, ZARA1 and ZARA2.

We first report on two key trajectory quality metrics: Maximum Mean Discrepancy (MMD) and Dynamic Time Warping (DTW). MMD is employed to assess the macroscopic distributional consistency between the predicted and ground-truth trajectories, while DTW excels at measuring the microscopic shape similarity between two trajectories and is robust to non-linear temporal variations. These two metrics serve as a powerful complement to the distance-based MAE and OT metrics in the main text, offering a more holistic view of the prediction fidelity. Subsequently, we conduct an in-depth comparison from the perspective of model efficiency, reporting two core metrics: the number of floating-point operations for a single forward pass (GFLOPs) and the continuous simulation frame rate (FPS). As a hardware-agnostic metric, GFLOPs reflects the theoretical computational complexity of the model's algorithm itself. In contrast, FPS directly measures the model's practical throughput and real-time performance on specific hardware.

As shown in Table 6 and Table 7, our method not only performs exceptionally on the trajectory quality metrics but also demonstrates a commanding advantage in runtime efficiency. These results further validate the superiority of our approach and its significant potential for practical applications.

Additionally, we evaluate the physical realism of our method using three metrics—Final Displacement Error (FDE), collision count (#Colli), and Density Estimation Accuracy (DEA)—across six datasets (GC, UCY, ETH, HOTEL, ZARA1 and ZARA2); the results are reported in Tables 8, 9, and 10, respectively. Across all datasets and metrics, our method consistently achieves the best performance, demonstrating superior physical plausibility, fewer inter-agent collisions, and more accurate crowd density estimation compared to all baselines.

Table 6: Additional performance comparison on GC and UCY datasets. The bold and underlined font show the best and the second best result respectively. Performance averaged over 5 runs.

| Model | GC | | | | UCY | | | |
|---|---|---|---|---|---|---|---|---|
| | MMD↓ | DTW↓ | GFLOPs↓ | FPS↑ | MMD↓ | DTW↓ | GFLOPs↓ | FPS↑ |
| CA | 0.0620 | - | - | - | 2.0220 | - | - | - |
| SFM | 0.0150 | - | - | - | 0.1209 | - | - | - |
| STGCNN | 0.5296 | 5.1438 | 0.0002 | 124.163 | 0.5149 | 5.1695 | 0.0002 | 124.163 |
| PECNet | 0.0397 | 0.7431 | 3.4704 | 18.9913 | 0.1504 | 2.0986 | 2.7134 | 19.0032 |
| MID | 0.3737 | 4.2773 | 0.4537 | 3.7826 | 0.4384 | 4.7109 | 0.3871 | 4.7312 |
| PCS | 0.0126 | 0.4378 | 0.8242 | 32.8106 | 0.1070 | 0.9887 | 0.2442 | 46.9936 |
| NSP | 0.0106 | **0.3329** | 0.2176 | 17.3792 | 0.1199 | 0.9965 | 0.2121 | 32.8713 |
| SPDiff | 0.0092 | 0.3332 | 0.1522 | 4.8315 | 0.0671 | 0.7541 | 0.1261 | 2.1247 |
| **Ours** | **0.0083** | 0.3441 | 0.1994 | 22.1406 | **0.0627** | **0.6905** | 0.0472 | 28.3033 |

## A.8 Computational cost

We provide a comparative analysis of our method against major baseline approaches in crowd simulation, focusing on three key metrics: training time, GPU memory consumption, and batch size, as summarized in the Table 11. Our method indeed incurs higher training costs due to: the node embedding matrix, which remains memory-intensive even under small batch sizes and the temporal coupling between consecutive frames, which requires storing intermediate tensors for flux computation and backpropagation through the ODE solver. However, this design is intentional: it ex-

Table 7: Additional performance comparison on ETH and HOTEL datasets. The bold and underlined font show the best and the second best result respectively. Performance averaged over 5 runs.

| Model | ETH | | | | HOTEL | | | |
|---|---|---|---|---|---|---|---|---|
| | MMD↓ | DTW↓ | GFLOPs↓ | FPS↑ | MMD↓ | DTW↓ | GFLOPs↓ | FPS↑ |
| CA | 0.4329 | - | - | - | 0.3548 | - | - | - |
| SFM | 0.3788 | - | - | - | 0.2301 | - | - | - |
| STGCNN | 3.9678 | 1.5622 | 0.0002 | 124.163 | 0.7418 | 3.7811 | 0.0002 | 124.163 |
| PECNet | 0.3573 | 0.3571 | 3.4704 | 19.9936 | 0.2219 | 0.2415 | 3.1927 | 30.3817 |
| MID | 3.7518 | 1.9512 | 0.5618 | 11.7528 | 0.6957 | 3.4641 | 0.4914 | 17.3411 |
| PCS | 0.3577 | 0.3101 | 0.3378 | 30.0575 | 0.1902 | 0.2201 | 0.2098 | 76.8331 |
| NSP | 0.3365 | 0.2843 | 0.2863 | 18.3782 | 0.1766 | 0.1883 | 0.1452 | 43.1812 |
| SPDiff | 0.2221 | 0.2275 | 0.1753 | 12.2830 | **0.1090** | **0.1444** | 0.0586 | 14.5837 |
| **Ours** | **0.1672** | **0.2221** | 0.2250 | 32.7108 | 0.1113 | 0.1464 | 0.0186 | 57.1481 |

Table 8: Additional performance comparison on GC and UCY datasets. The bold and underlined font show the best and the second best result respectively. Performance averaged over 5 runs.

| Model | GC | | | UCY | | |
|---|---|---|---|---|---|---|
| | FDE↓ | #Colli↓ | DEA↓ | FDE↓ | #Colli↓ | DEA↓ |
| CA | 4.937 | 638 | 0.0290 | 13.9788 | 4602 | **0.0266** |
| SFM | 2.1031 | 1368 | 0.0280 | 3.4871 | 554 | 0.0267 |
| STGCNN | 15.3913 | >9999 | 0.1867 | 10.4219 | >9999 | 0.3123 |
| PECNet | 3.8124 | 1593 | 0.0612 | 5.9714 | 1474 | 0.0296 |
| MID | 13.8824 | >9999 | 0.1323 | 13.9821 | >9999 | 0.2134 |
| PCS | 1.9742 | 558 | 0.0271 | 3.0814 | 474 | 0.0329 |
| NSP | 1.4124 | 1006 | 0.0265 | 3.4781 | 438 | 0.0332 |
| SPDiff | 1.3842 | 944 | 0.0248 | 3.0714 | 738 | 0.0467 |
| **Ours** | **1.3628** | **492** | **0.0231** | **2.9413** | **432** | 0.0337 |

plicitly enforces spatiotemporal continuity constraints, enabling high-fidelity, physically consistent predictions at inference time. As demonstrated in our experiments, our approach achieves superior trajectory plausibility and density conservation while maintaining competitive inference speed. In essence, we trade a moderate increase in training overhead for stronger physical inductive biases and improved generation quality.

## A.9 CASE STUDY

To provide an intuitive demonstration of our method's performance and to complement the quantitative comparisons, we visualized trajectory prediction results of our proposed method and physics-guided crowd simulation methods on four datasets, as shown in Figures 5. It can be observed that all methods fit the ground-truth trajectories well in the short term. However, for longer-term predictions, other methods tend to deviate from the true paths, while STDDN remains the closest to the ground truth. Moreover, in the leftmost example, it can be seen that part of PCS's predicted trajectory overlaps with obstacles, which is clearly an unreasonable prediction.

## A.10 ACCUMULATED DENSITY PREDICTION ERROR ANALYSIS

To comprehensively evaluate the density prediction performance of the proposed method over long time frames, we analyzed the trend of the average absolute error (MAE) between predicted and true densities over time on the GC and UCY datasets. We compared the results with two representative methods in crowd simulation, PCS and SPDiff. The experimental results are shown in Figure 6 and Figure 7. From Figure 6 and 7, it can be seen that our proposed method consistently maintains the lowest overall density prediction error. In contrast, while the SPDiff method is capable of

Table 9: Additional performance comparison on ETH and HOTEL datasets. The bold and underlined font show the best and the second best result respectively. Performance averaged over 5 runs.

| Model | ETH | | | HOTEL | | |
|---|---|---|---|---|---|---|
| | FDE↓ | #Colli↓ | DEA↓ | FDE↓ | #Colli↓ | DEA↓ |
| CA | 1.2343 | 738 | 0.0184 | 0.9641 | 714 | 0.0326 |
| SFM | 1.1489 | 682 | 0.0184 | 0.8731 | 630 | 0.0328 |
| STGCNN | 5.9631 | 2410 | 0.1323 | 8.7814 | 2031 | 0.1322 |
| PECNet | 0.9412 | 780 | **0.0156** | 0.8104 | 1508 | **0.0320** |
| MID | 6.0431 | 3814 | 0.1321 | 7.9873 | 1873 | 0.1211 |
| PCS | 0.9514 | 692 | 0.0240 | 0.7641 | 634 | 0.0332 |
| NSP | 0.9312 | 628 | 0.0243 | 0.6971 | 591 | 0.0341 |
| SPDiff | 0.8536 | 700 | 0.0218 | 0.5931 | 608 | 0.0345 |
| **Ours** | **0.8391** | **596** | 0.0251 | **0.5841** | **544** | 0.0339 |

Table 10: Additional performance comparison on ZARA1 and ZARA2 datasets. The bold and underlined font show the best and the second best result respectively. Performance averaged over 5 runs.

| Model | ZARA1 | | | | ZARA2 | | | |
|---|---|---|---|---|---|---|---|---|
| | FDE↓ | MAE↓ | #Colli↓ | DEA↓ | FDE↓ | MAE↓ | #Colli↓ | DEA↓ |
| CA | 2.0341 | 1.2814 | 262 | 0.0178 | 1.9432 | 1.0311 | 4031 | 0.0093 |
| SFM | 1.9857 | 1.1446 | 262 | 0.0171 | 2.9841 | 1.7120 | 3146 | 0.0087 |
| STGCNN | 6.9871 | 3.7361 | 2085 | 0.1341 | 8.7638 | 4.1324 | 7113 | 0.2141 |
| PECNet | 1.5314 | 0.9848 | 244 | 0.0162 | 3.9436 | 2.0405 | 4656 | 0.0192 |
| MID | 7.9981 | 4.0312 | 3013 | 0.1492 | 9.9743 | 5.4327 | 6016 | 0.1934 |
| PCS | 1.3289 | 0.8393 | 236 | 0.0157 | 2.8377 | 1.1837 | 3686 | **0.0072** |
| NSP | 1.4312 | 0.8241 | 226 | **0.0147** | 2.1842 | 1.1717 | 3686 | 0.0083 |
| SPDiff | 1.1244 | 0.7450 | 246 | 0.0204 | 1.9811 | 0.9327 | 3021 | 0.0091 |
| **Ours** | **1.1132** | **0.7338** | **220** | 0.0150 | **1.8321** | **0.9000** | **2920** | 0.0086 |

predicting density to some extent, its performance does not significantly outperform other methods. Our method, however, takes the density prediction problem into full consideration and effectively embeds physical laws into the prediction model. Therefore, even without achieving the highest precision in density prediction, our method significantly improves the effectiveness of long-duration crowd simulation.

## A.11 SENSITIVITY STUDY

We conducted a sensitivity analysis of the key hyperparameters in the framework across multiple datasets, including grid size, ODE time steps ($\tau$), balance coefficients of the loss function, and node embedding dimensions. Figure 8, 9, 10 shows the specific performance of the model in terms of MAE on the UCY, ETH, and HOTEL datasets. By examining Figure 8, 9, and 10, it can be observed that the grid size affects the simulation performance to some extent in all three datasets. Only by selecting an appropriate grid size can the simulation performance of the microscopic network reach its optimal level. Specifically, the grid size influences the pedestrian flow rate, and an appropriate grid size optimizes the constraint effect of the continuity equation; the ODE step size affects the degree of overfitting in the network during simulation extrapolation, and an optimal step size ensures the best performance in long-term simulations; the balance coefficient in the loss function determines the performance balance between the microscopic trajectory prediction network and the macroscopic density prediction, and a reasonable balance can achieve the best microscopic prediction performance; the grid node embedding dimension affects the number of parameters in the neural ODE, and too many parameters can lead to overfitting, thus limiting the constraint effect of the continuity equation on the trajectory prediction network.

Table 11: Computational cost compared to other methods on GC and UCY datasets.

| Model | GC | | | UCY | | |
|---|---|---|---|---|---|---|
| | Training time | Memory | Batch size | Training time | Memory | Batch size |
| PCS | 1h | 2G | 32 | 1h | 2G | 16 |
| SPDiff | 5h | 7G | 32 | 3h | 6G | 32 |
| **Ours** | 8h | 26G | 16 | 5h | 22G | 4 |

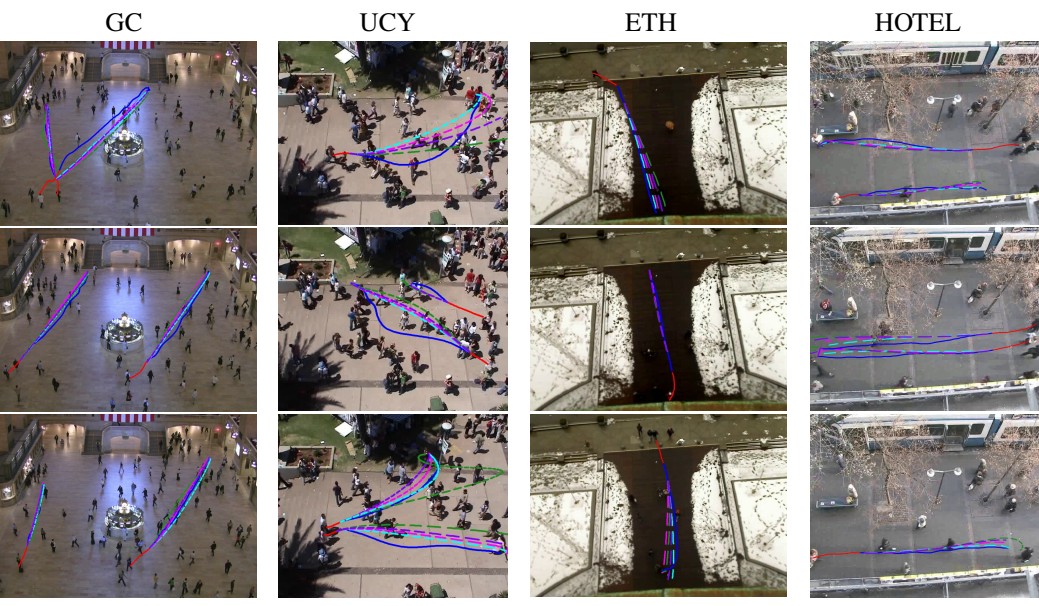

Figure 5: Visualization of predicted trajectories on the GC, UCY, ETH, and HOTEL datasets. Each column represents a different dataset, showcasing various scenarios. Observed trajectories are shown in ——, ground truth in ——, STDDN in - - -, SPDiff in - - -, NSP in - - -, and PCS in - - -.

## A.12 BOUNDARY EFFECTS

In our current implementation, we approximate boundary effects through dynamic population handling: at each time step, we only model pedestrians currently visible in the scene. Individuals entering or exiting the field of view are explicitly detected and their states are initialized or terminated accordingly. This strategy has proven effective in practice across standard benchmarks, where entry/exit events are relatively sparse and well-separated. However, we fully acknowledge the theoretical limitation: in truly open environments, future entries cannot be predicted a priori, and strict mass conservation over a fixed spatial domain is indeed violated. A more principled solution would be to extend the continuity equation with explicit source/sink terms:

$$\frac{\partial \rho}{\partial t} + \nabla \cdot (\rho \mathbf{v}) = S \tag{22}$$

where $S > 0$ models inflow (new entrants) and $S < 0$ models outflow (exits). Such an extension would not only better reflect open-system dynamics but also enable prediction of potential entry zones—e.g., via learned or context-aware source maps. We agree this is a highly promising direction and plan to explore it thoroughly in future work.

## A.13 LIMITATIONS

Despite the promising results achieved in this study, several limitations remain, which also point to promising directions for future research. First, in terms of spatial modeling, future work could explore more flexible representations, such as implicit function modeling based on continuous spatial

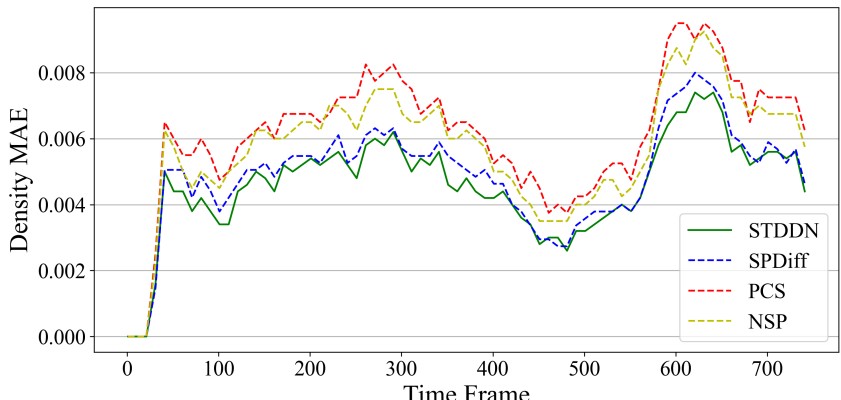

Figure 6: Accumulated density prediction error comparison on the GC dataset.

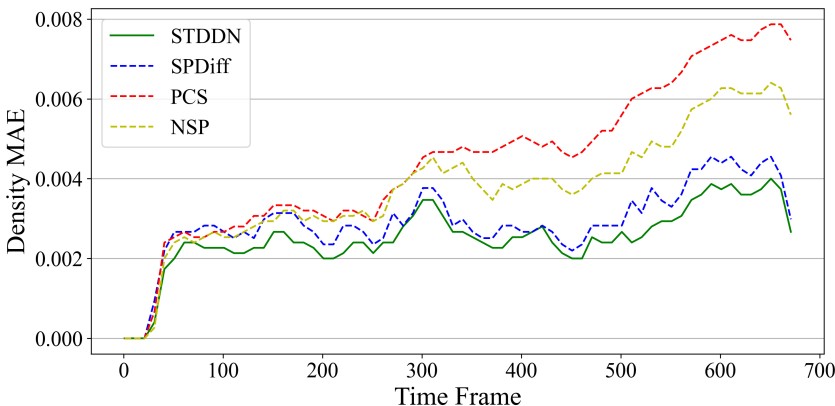

Figure 7: Accumulated density prediction error comparison on the UCY dataset.

coordinates, or multi-scale graph structures to alleviate the computational overhead and expressive limitations introduced by grid discretization. Such approaches may better capture complex spatial dependencies that extend beyond grid partitions. Second, regarding the use of the continuity equation as a strong physical constraint—which, while enhancing consistency and stability, may override certain latent motion patterns—future research could investigate soft-constraint mechanisms or adaptive strategies that balance data-driven and physics-driven modeling. This would preserve physical consistency while improving the model's ability to capture hidden dynamics in the data.

In addition, enhancing computational efficiency in large-scale scenarios is another important direction, for instance through sparsification techniques, model compression, or parallel acceleration, to further facilitate the practical deployment of the proposed framework in crowd simulation, emergency evacuation, and intelligent transportation applications.

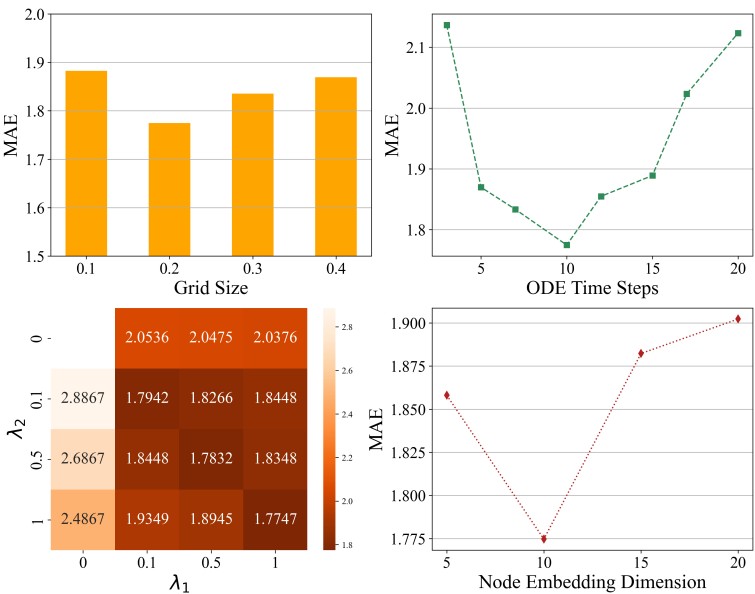

Figure 8: Sensitivity analysis on the UCY dataset for grid size, ODE time steps $\tau$, loss balancing coefficients ($\lambda_1$ and $\lambda_2$), and node embedding dimension, using MAE as the evaluation metric.

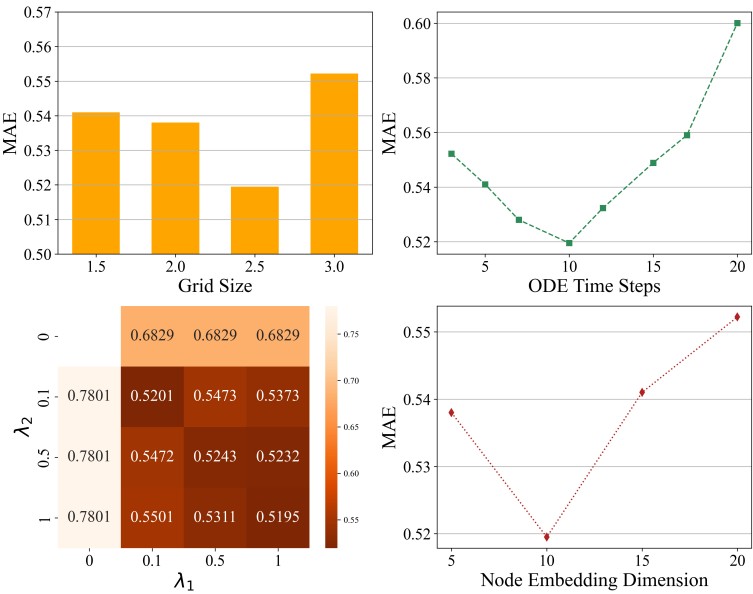

Figure 9: Sensitivity analysis on the ETH dataset for grid size, ODE time steps $\tau$, loss balancing coefficients ($\lambda_1$ and $\lambda_2$), and node embedding dimension, using MAE as the evaluation metric.

## A.14 USAGE OF LLMS

All aspects of this work, including the conceptualization, methodology, experiments, and analysis, were independently carried out by the author. The final manuscript was refined with the assistance of a large language model for language polishing to improve clarity and coherence. However, the intellectual content and scientific contributions remain solely the work of the author.

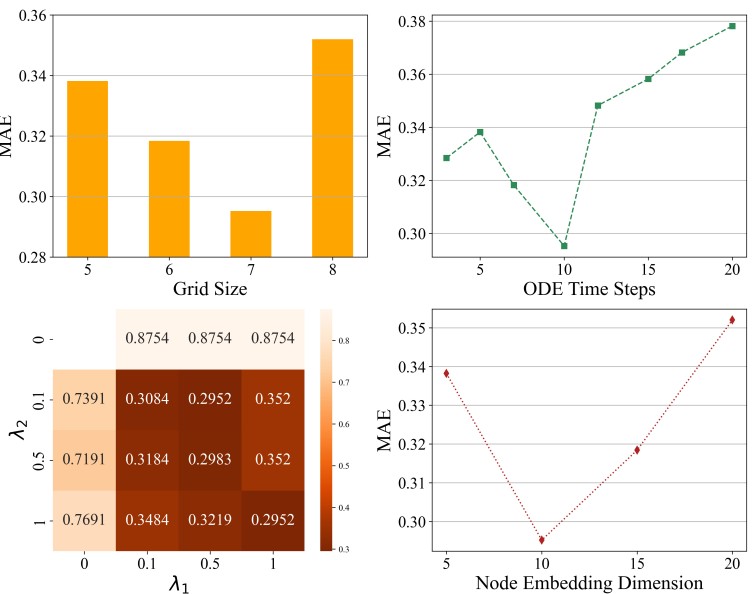

Figure 10: Sensitivity analysis on the HOTEL dataset for grid size, ODE time steps $\tau$, loss balancing coefficients ($\lambda_1$ and $\lambda_2$), and node embedding dimension, using MAE as the evaluation metric.

