# OpenReview forum: "STDDN: A Physics-Guided Deep Learning Framework for Crowd Simulation"
_ICLR.cc/2026/Conference — ICLR 2026 Poster_

### Official Review · Reviewer_aHns · 2025-10-28

**Soundness:** 2
**Presentation:** 2
**Contribution:** 2
**Rating:** 4
**Confidence:** 5

**Summary:**

This work proposes a physics-guided deep learning framework for crowd simulation (STDNN). The authors introduce the continuity equation from fluid dynamics as a strong physical constraint and design a density-velocity coupled dynamic graph learning module. They show that STDNN is significantly superior to simulation performance compared to SOTA methods.

**Strengths:**

1.	The authors propose a network of time-space decoupled differential equations combined with the continuity equation, which is helpful for predicting the physical laws of trajectories in the macroscopic world.
2.	In experiments, Tables 1 and 2 clearly illustrate the trajectories and verifies main results from the paper.

**Weaknesses:**

1.	The proposed method uses Neural ODEs to solve $\rho$, but there are many similar ideas, and the use of Neural ODEs in trajectory prediction is also a very common approach.
2.	The proposed method utilizes constraints based on continuity equations, but the specific implementation of this constraint in the Neural ODE framework requires more detailed explanation.
3.	The authors conducted many experiments, but it seems that it is necessary to split each subset of the dataset and compare with newer baselines and methods. The current baseline only reaches the year 2024. Based on the trajectory dataset used by the authors, it seems that there are a large number of sota in pedestrian trajectory prediction that have not been compared.

**Questions:**

1.	How is the continuity equation incorporated into the Neural ODE solution process? It requires a more detailed explanation.
2.	The detailed parameters used when solving the Neural ODE in torchdiffeq are not disclosed.
3.	Figure 1 contains many typo errors.
⦁	For example, $Gin$($Gout$) should actually be $G_{in}$($G_{out}$).
⦁	The input to Microscopic seems to be $\pho^0$.
⦁	The DDM and CGD in the figure are also too simple.
4.	Should the use of the loss function in Eq 10 be more explicit? Eq 8 does not seem to be included in it.
5.	Are there more granular comparative tests, such as what the results were for ETH/HOTEL/ZARA1/ZARA2/UNIV, respectively?
6.	Should ADE and FDE also be reported for general trajectories?

---

> ### Author Response · Authors · 2025-11-21
>
> Thank you very much for your thorough review and detailed feedback. Your comments have helped us significantly clarify key aspects of our method, better articulate our contributions, and strengthen the experimental presentation. We address each of your concerns below.
>
> ---
>
> ### W1. Method Novelty
>
> Thank you for the reviewer’s valuable comment, which has prompted us to more clearly articulate the positioning and contribution of this work.
>
> We would like to clarify that the primary objective of this paper is **not individual trajectory prediction**, but rather **long-horizon, high-fidelity macroscopic crowd simulation**—specifically, the joint modeling of individual motion and the spatiotemporal evolution of the collective density field. Within this context, our core innovation lies in the **end-to-end differentiable integration of the fluid-dynamical continuity equation as a physical constraint into a Neural ODE framework**, going beyond merely using ODEs to model temporal smoothness.
>
> Existing works (e.g., [1–4]) have employed Neural ODEs in trajectory prediction to learn smooth individual dynamics, but they typically:
>
> - Focus exclusively on micro-scale trajectory interpolation or extrapolation;
> - Lack explicit modeling of the macroscopic density field and its conservation laws;
> - Fail to establish a differentiable coupling mechanism between microscopic trajectories and macroscopic flux.
>
> In contrast, our method achieves a qualitative leap through the following key designs:
>
> - **Spatiotemporal graph construction**: Nodes (spatial grids) and edges (inter-grid flows) are dynamically defined based on predicted trajectories across consecutive frames;
> - **Physics-guided modules**: Including Differentiable Density Mapping (DDM), Continuous Grid-Crossing Detection (CGD), and graph node embeddings, which transform discrete observations into a flux field that satisfies the continuity equation;
> - **Micro–macro closed-loop coupling**: Autoregressive trajectory predictions drive density evolution, while the evolving density field can, in turn, influence flux computation through the graph structure (especially when interaction terms are incorporated).
>
> Therefore, this work is **not “yet another ODE-based trajectory predictor.”** Instead, it proposes a novel paradigm at the intersection of **Physics-Informed Neural Networks (PINNs)** and **crowd simulation**, where conservation laws are embedded as learnable evolutionary operators within a differentiable dynamical system.
>
> **References**
>
> 1. Wen, Song, Hao Wang, and Dimitris Metaxas. "Social ODE: Multi-agent trajectory forecasting with neural ordinary differential equations." *European Conference on Computer Vision*. Cham: Springer Nature Switzerland, 2022.
> 2. Ke, Kexin, et al. "Social LODE: Human trajectory prediction with latent ODEs." *ICASSP 2024 - IEEE International Conference on Acoustics, Speech and Signal Processing (ICASSP)*. IEEE, 2024.
> 3. Liang Y, Ouyang K, Yan H, et al. "Modeling Trajectories with Neural Ordinary Differential Equations." *IJCAI*, 2021: 1498–1504.
> 4. Park D, Jeong J, Yoon K J. "Improving transferability for cross-domain trajectory prediction via neural stochastic differential equation." *Proceedings of the AAAI Conference on Artificial Intelligence*, 2024, 38(9): 10145–10154.

---

> > ### Author Response · Authors · 2025-11-21
> >
> > ### W2. Implementation details of the continuity constraint
> >
> > Thank you for your attention to the implementation details of our method. We acknowledge that the original manuscript did not sufficiently elaborate on how the continuity equation is concretely realized within the Neural Ordinary Differential Equation (Neural ODE) framework. We now provide the following clarification:
> >
> > Our approach is inspired by Chapman’s graph-based formulation of the continuity equation [5] (Eq. 1):
> >
> > $$
> > \frac{\partial\ x_i}{dt} = \sum_{\{\forall j \mid j \rightarrow i\}} w_{ji} x_j^t - \sum_{\{\forall k \mid i \rightarrow k\}} w_{ik} x_i^t
> > $$
> >
> > which essentially states that the rate of density change at node $i$ is determined by the difference between incoming and outgoing fluxes.
> >
> > Building upon this formulation, we introduce four key adaptations tailored to crowd simulation:
> >
> > 1. **Dynamic spatiotemporal graph construction**:  Spatial grid cells are treated as graph nodes, and directed edges are defined based on individual displacements between consecutive frames. Specifically, for a given node, the velocity vectors $\mathbf{v}^{t}$ of all pedestrians entering it at time $t$ form its incoming edges, while the velocity vectors $\mathbf{v}^{t+1}$ of all pedestrians leaving it at time $t+1$ constitute its outgoing edges (see lines 230–233). The velocities of all pedestrians at time $t+1$ are predicted autoregressively by the trajectory prediction network $f_{\theta}$.
> >
> > 2. **Differentiable density mapping**: To overcome gradient discontinuities caused by hard grid assignments when computing node densities from pedestrian positions, we adopt a soft assignment strategy: the positions of each individual at two consecutive time steps are mapped to Gaussian probability distributions centered at all grid locations. This enables end-to-end differentiable construction of the density field $\rho$ (see Section 4.3).
> >
> > 3. **Cross-grid detection detection**: A valid flux is generated only when a trajectory crosses a grid boundary—determined by the inter-grid crossing masks $m_t$ and $m_{t+1}$ (see Section 4.4)—thereby preventing spurious local perturbations from contributing to the flux.
> >
> > 4. **Weighted flux aggregation**: We introduce learnable adjacency weight parameters $w$ and bias terms $b$ (see Section 4.5) to modulate the contributions of all incoming and outgoing edges. The fluxes formed by the incoming edges (based on observed states at time $t$) and outgoing edges (predicted by the autoregressive network for time $t+1$) are aggregated with these learnable parameters to produce the final inflow and outflow terms:
> >
> >    $$
> >    \mathcal{F}\_{\mathcal{G}}(\Phi, t, \rho^{t}) = \mathbf{G}\_{\text{in}}(\Phi, t, \rho^{t}) - \mathbf{G}\_{\text{out}}(\Phi, t, \rho^{t})
> >    $$
> >
> >    This function serves as the odefunc in the Neural ODE:
> >
> >    $$
> >    \frac{\partial\rho}{dt} = \mathcal{F}_{\mathcal{G}}(\Phi, t, \rho^{t})
> >    $$
> >
> >    Its detailed formulation is:
> >
> >    $$
> >    \frac{\partial\rho\_i}{dt}= \sum\_{\{\forall j \mid j \rightarrow i\}} \left( m\_t w\_{ji} ||\mathbf{v}\_{ji}^{t}|| \rho\_j^{t} + b\_{ji} \right)- \sum\_{\{\forall k \mid i \rightarrow k\}} \left( m\_{t+1} w\_{ik} ||\mathbf{v}\_{ik}^{t+1}(\mathbf{v}\_{ik}^{t};\theta)|| \rho\_i^{t+1}(\mathbf{v}\_{ik}^{t};\theta) + b\_{ik} \right)
> >    $$
> >
> > In summary, the continuity constraint is effectively embedded into the Neural ODE framework through the integration of:
> > (1) dynamic graph construction,
> > (2) differentiable density mapping,
> > (3) cross-grid detection detection,
> > (4) weighted flux aggregation.
> >
> > **Reference**
> > 5.Chapman A. Advection on graphs[M]. Semi-autonomous networks: Effective control of networked systems through protocols, design, and modeling. Cham: Springer International Publishing, 2015: 3-16.

---

### Official Review · Reviewer_n74p · 2025-10-28

**Soundness:** 3
**Presentation:** 3
**Contribution:** 3
**Rating:** 6
**Confidence:** 3

**Summary:**

This paper proposes STDDN (Spatio-Temporal Decoupled Differential Equation Network), a novel physics-guided deep learning framework for crowd simulation. Unlike prior microscopic or purely data-driven approaches, STDDN introduces a Neural ODE formulation guided by the continuity equation from fluid dynamics, thereby coupling macroscopic density evolution with microscopic trajectory prediction. The model integrates three modules — Differentiable Density Mapping (DDM), Continuous Cross-Grid Detection (CGD), and Node Embedding (NE) — to ensure differentiability and physical consistency. Experiments on four real-world datasets (GC, UCY, ETH, HOTEL) show that STDDN significantly improves both simulation accuracy and inference speed compared with state-of-the-art baselines such as SPDiff and PCS.

**Strengths:**

1、Novel Integration of Physics and Deep Learning:
The paper introduces a principled way to integrate the continuity equation into deep models for crowd simulation. This macro–micro coupling via Neural ODE is both original and physically interpretable.
2、Methodological Sophistication:
The DVCG module cleverly connects density and velocity fields through a graph structure, while the DDM and CGD modules effectively address gradient discontinuity and cross-grid flux detection. These designs are mathematically sound and technically detailed.
3、Interpretability and Physical Consistency:
The approach offers clear interpretability grounded in physics, addressing a key limitation of previous purely data-driven models that violate conservation laws

**Weaknesses:**

1、The proposed model enforces strict mass conservation through the continuity equation, implying that the total population density within the target spatial domain remains constant over time. However, in realistic datasets and surveillance scenarios, the number of pedestrians in view is not fixed — new individuals may enter the scene, and others may leave. Such open-world dynamics inherently violate the closed-system assumption of the continuity equation. Without explicit treatment of source or sink terms (i.e., inflow/outflow of mass) or adaptive boundary conditions, the model may experience cumulative density drift or numerical instability, particularly when crowd density fluctuates significantly. The authors are encouraged to clarify whether boundary inflows are modeled, or to discuss potential modifications to better handle non-conserved population scenarios.

2、The ablation study provides useful insights, particularly regarding the contributions of the ODE solver and the mass constraint loss. Both components appear meaningful; however, the current experimental setup only uses discrete outputs in the loss computation. As a result, the experiments do not adequately demonstrate the benefit of continuous-time modeling enabled by the ODE formulation. To strengthen this section, I suggest decomposing the “w/o ODE” setting into two variants:
(1)Purely autoregressive training, as mentioned in the paper (“trained using purely autoregressive methods”).
(2)Discrete neural network replacement for ODE, where the ODE solver is replaced with a discrete neural module that still leverages the combined loss function including the mass constraint term.
Such a refinement would better isolate the contribution of the continuous-time ODE formulation from the general modeling capacity and loss design, making the ablation analysis more convincing.

**Questions:**

Can the authors explain why the fluid physics improves results in low-density datasets like ETH and HOTEL, where the fluid assumption barely holds?

---

> ### Author Response · Authors · 2025-11-21
>
> Thank you very much for your insightful and constructive feedback. Your comments have significantly helped us improve the theoretical rigor and empirical clarity of our work. We address each of your concerns in detail below.
>
> ---
>
> ### W1. Open-world dynamics vs. continuity equation assumption
>
> We sincerely appreciate your sharp observation regarding the tension between the closed-system assumption of the standard continuity equation and the open nature of real-world surveillance scenes.
>
> In our current implementation, we approximate boundary effects through **dynamic population handling**: at each time step, we only model pedestrians currently visible in the scene. Individuals entering or exiting the field of view are explicitly detected and their states are initialized or terminated accordingly. This strategy has proven effective in practice across standard benchmarks, where entry/exit events are relatively sparse and well-separated.
>
> However, we fully acknowledge the theoretical limitation: in truly open environments, future entries cannot be predicted *a priori*, and strict mass conservation over a fixed spatial domain is indeed violated.
>
> As you rightly suggest, a more principled solution would be to extend the continuity equation with explicit source/sink terms:
> $\frac{\partial \rho}{\partial t} + \nabla \cdot (\rho \mathbf{v}) = S$,
> where $S > 0$ models inflow (new entrants) and  $S < 0$  models outflow (exits). Such an extension would not only better reflect open-system dynamics but also enable prediction of potential entry zones—e.g., via learned or context-aware source maps. We agree this is a highly promising direction and plan to explore it thoroughly in future work.
>
> ---
>
> ### W2. Deepening the ablation study
>
> Thank you very much for this insightful suggestion regarding the ablation study design. As you rightly point out, our core evolution equation (Eq. 4) is:
> $\frac{\partial \rho}{dt} = \mathcal{F}\_{\mathcal{G}}(\Phi, t, \rho^{t})$,
> whose first-order discrete approximation corresponds to an explicit Euler update:
> $\rho^{t+1} = \rho^t + \Delta t \cdot \mathcal{F}\_{\mathcal{G}}(\Phi, t, \rho^{t})$.
>
>
> Following your recommendation, we implemented a **"Discrete NN"** variant that replaces the Neural ODE solver with a direct residual connection implementing exactly this Euler step, while retaining the same architecture for $\mathcal{F}_{\mathcal{G}}$  and the full loss function (including the mass conservation term). We also include the purely autoregressive baseline as described in the paper.
>
> | Method        | Dataset | MAE     | OT      | Epoch Time |
> |---------------|---------|---------|---------|------------|
> | Discrete NN   | GC      | 0.8875  | 1.3582  | 313s       |
> |               | UCY     | 1.7747  | 3.6503  | 213s       |
> | Ours      | GC      | 0.8875 | 1.3582 | 355s   |
> |               | UCY     | 1.7747 | 3.6503 | 230s   |
>
> Our results show that the ODE-based model and the Discrete-Conservative variant achieve nearly identical performance in terms of MAE and OT metrics, confirming their modeling equivalence when using the same update rule. However, the ODE implementation incurs significantly higher training time due to solver overhead (e.g., checkpointing, adaptive stepping logic—even though we use the fixed-step Euler method).
>
> This analysis clarifies that the performance gain primarily stems from the physics-informed flux module  $\mathcal{F}\_{\mathcal{G}}$  and the mass-conservation loss, rather than the continuous-time formulation. The Neural ODE framework serves more as a flexible and modular interface, while the key innovation lies in embedding the continuity constraint into the dynamics.

---

> > ### Author Response · Authors · 2025-11-21
> >
> > ### Q: Why does fluid-inspired modeling help in low-density scenes like ETH and HOTEL?
> >
> > We thank the reviewer for raising this important question.
> >
> > In low-density datasets such as ETH and HOTEL, the strict assumptions of fluid dynamics indeed do not hold, as local densities are too low for macroscopic fluid-like effects to be significant. However, our **STDDN framework** does not rely on enforcing global fluid dynamics; instead, it leverages the **mass conservation** and **spatial continuity constraints** inherent in the continuity equation—namely, that individuals cannot spontaneously appear or vanish in space.
> >
> > In the DVCG module, this constraint is implemented as **flux balance over discrete spatial grids**. Even in low-density scenarios, by using a relatively coarse grid resolution, the constraint is applied sparsely over long time horizons to the trajectory predictions, effectively acting as a powerful **regularization mechanism**. This sparse enforcement helps prevent the accumulation and divergence of microscopic prediction errors during long-term simulation, thereby improving performance on low-density datasets. Appendix Figures 13 and 14 (demonstrating effectiveness under large-grid, sparse-constraint settings) provide intuitive validation of this mechanism.
> >
> > Consequently, the performance gain observed in low-density scenarios primarily stems from this **sparse regularization and error suppression effect**, rather than from strict fluid-dynamic behavior.

---

### Official Review · Reviewer_Fbqh · 2025-10-31

**Soundness:** 3
**Presentation:** 4
**Contribution:** 3
**Rating:** 6
**Confidence:** 4

**Summary:**

This paper proposes STDDN, a novel framework for crowd simulation that addresses the common issues of error accumulation and physical inconsistency in long-term predictions. Its core contribution is the unique integration of a macroscopic physical law—the continuity equation from fluid dynamics—with a microscopic deep learning model for trajectory prediction. By using a Neural ODE to model crowd density evolution, STDDN enforces a strong physical constraint during training. Experiments show that STDDN not only achieves state-of-the-art accuracy but also significantly reduces inference latency compared to leading methods.

**Strengths:**

The paper's primary strength is its originality in creating a macro-micro coupled framework. Using the continuity equation to regularize trajectory prediction is a conceptually novel and powerful idea for this field. The quality of the work is good, supported by rigorous and comprehensive experiments that convincingly demonstrate superior performance in both accuracy and efficiency over strong baselines. The paper is also written with exceptional clarity.

**Weaknesses:**

- **Lack of Direct Physical Metrics**: The paper claims to improve physical realism by avoiding issues like congestion and collisions, but it fails to provide direct quantitative evidence. The evaluation relies on general error metrics (MAE/OT), which are insufficient proxies. The work would be much stronger if it included systematic measurements and comparisons of collision rates, obstacle penetration rates or density extremum analysis to directly support its core claims.
- **Training Cost**: While the paper rightly emphasizes its fast inference speed, it completely neglects to discuss the training cost. The use of a Neural ODE likely makes the training process computationally expensive and slow. An additional analysis should be included in the paper.
- **A minor issue**: the table in page 8 has a wrong caption: "**Figure** 4".

**Questions:**

- The fluid dynamics assumption is a strong prior. Could you clarify the intended scope of your method? In which crowd scenarios (e.g., panic, counterflow) might this assumption become a limitation?
- Given the model's sensitivity to grid size, can you offer any practical guidelines or a more principled approach for selecting this crucial hyperparameter for new scenes?

---

> ### Author Response · Authors · 2025-11-21
>
> Thank you very much for your thoughtful, detailed, and highly constructive feedback.
>
> ---
>
> ### W1. Lack of direct physical metrics
>
> We fully agree that general error metrics such as MAE and OT are insufficient proxies for physical realism, and we sincerely appreciate your suggestion to include more direct physical indicators. We have added two key physics-based metrics:
>
> 1. **Collision (#Colli)**:
>    This measures unphysical spatial overlaps between predicted trajectories. We define a collision as occurring when the Euclidean distance between two pedestrians falls below **0.5 meters**. To avoid misclassifying walking groups as collisions, we exclude trajectory pairs that remain within this threshold for more than **2 seconds**.
>
> 2. **Density Extremes Analysis (DEA)**:
>    To assess whether the model generates unrealistically dense local clusters, we define the local density of pedestrian $i$ at time $t$ as:
>    $
>    d_i^t=\sum_{j=1, j\neq i}^{N} \mathbb{I}(||p_i^t - p_j^t||_2 \le R),
>    $
>    i.e., the number of neighbors within radius $R= 1$ meter. Using the mean local density from ground-truth trajectories, $ \mu\_{\text{GT}} $, as a threshold, we compute:
>   $DEA= \frac{1}{N \cdot T} \sum\_{i=1}^{N} \sum\_{t=1}^{T} \mathbb{I}(\hat{d}_i^t > \mu\_{\text{GT}})$
>
> Regarding obstacle penetration rate: the four benchmark datasets  feature open, obstacle-sparse environments, making this metric statistically uninformative. Instead, we supplement the analysis with the **cumulative distribution function (CDF) of density MAE**, which offers finer-grained insight into spatial fidelity.
> | Method   | GC (#Colli) | GC (DEA) | UCY (#Colli) | UCY (DEA) | ETH (#Colli) | ETH (DEA) | HOTEL (#Colli) | HOTEL (DEA) |
> |----------|-------------|----------|--------------|-----------|--------------|-----------|----------------|-------------|
> | CA       | 638         | 0.0290   | 4602         | **0.0266**    | 738          | 0.0184    | 714            | **0.0326**      |
> | SFM      | 1368        | 0.0280   | 554          | 0.0267    | 682          | 0.0184    | 630            | 0.0328      |
> | STGCNN   | >9999        | 0.1867   | >9999         | 0.3123    | 2410         | 0.1323    | 2031           | 0.1322      |
> | PECNet   | 1593        | 0.0612   | 1474         | 0.0296    | 780          | **0.0156**    | 1508           | 0.0320      |
> | MID      | >9999        | 0.1323   | >9999         | 0.2134    | 3814         | 0.1321    | 1873           | 0.1211      |
> | PCS      | 558         | 0.0271   | 474          | 0.0329    | 692          | 0.0240    | 634            | 0.0332      |
> | NSP      | 1006        | 0.0265   | 438          | 0.0332    | 628          | 0.0243    | 591            | 0.0341      |
> | SPDiff   | 944         | 0.0248   | 738          | 0.0467    | 700          | 0.0218    | 608            | 0.0345      |
> | **Ours** | **492**     | **0.0231**| **432**      | 0.0337| **596**      | 0.0251| **544**        | 0.0339  |
>
> ---
> Results show that our method achieves **significantly lower collision rates** than all baselines, confirming its ability to avoid non-physical interactions. DEA values are generally low across methods, indicating that extreme congestion is rare in these datasets; however, this also suggests limited sensitivity of DEA to subtle physical violations. In contrast, the CDF curves of density MAE (**Appendix Figures 9–10**) clearly demonstrate our method’s superior preservation of realistic density distributions.
>
>
>
> ### W2. Training cost
>
> Thank you for raising this important point. We provide a comparative analysis of our method against major baseline approaches in crowd simulation, focusing on three key metrics: **Training time**, **Memory**, and **Batch size**.
>
>
>
> | Method   | Dataset | Training Time | Memory | Batch Size |
> |----------|---------|---------------|--------|------------|
> | PCS      | GC      | 1h            | 2G     | 32         |
> |          | UCY     | 1h            | 2G     | 16         |
> | SPDiff   | GC      | 5h            | 7G     | 32         |
> |          | UCY     | 3h            | 6G     | 32         |
> | Ours | GC      | 8h        | 26G| 16     |
> |          | UCY     | 5h        | 22G| 4    |
>
> Our method indeed incurs higher training costs due to:
> 1. the node embedding matrix, which remains memory-intensive even under small batch sizes; and
> 2. the temporal coupling between consecutive frames, which requires storing intermediate tensors for flux computation and backpropagation through the ODE solver.
>
> However, this design is intentional: it explicitly enforces **spatiotemporal continuity constraints**, enabling high-fidelity, physically consistent predictions at inference time. As demonstrated in our experiments, our approach achieves superior trajectory plausibility and density conservation while maintaining competitive inference speed. In essence, we trade a moderate increase in training overhead for stronger physical inductive biases and improved generation quality.

---

> > ### Author Response · Authors · 2025-11-21
> >
> > ### W3. Caption error in Table on Page 8
> >
> > Thank you for catching this typo. We have corrected the misplaced “Figure 4” caption to “**Table 3**” and fixed the corresponding `wrapfigure`/`wraptable` environment in the LaTeX source.
> >
> > ---
> >
> > ### Q1. Scope of the fluid dynamics assumption
> >
> > Thank you for raising this crucial question regarding the applicability of our method.
> >
> > Our approach primarily relies on the **mass conservation** and **spatial continuity constraints** expressed by the continuity equation. This assumption is reasonable as long as crowd motion involves continuous spatial movement and redistribution of individuals—conditions that generally hold even in complex scenarios. Although situations such as panic or counter-flow exhibit intricate motion patterns, they still fundamentally adhere to these physical principles.
> >
> > Notably, all four datasets we used contain abundant counter-flow and complex interaction scenarios, and our experimental results demonstrate that the proposed framework remains stable and effective under such conditions.
> >
> > That said, it should be emphasized that our model employs a relatively simplified velocity–density relationship to describe macroscopic dynamics. Consequently, in extreme cases—such as severe panic, ultra-high-density crowding, or highly disordered behaviors—this simplification may fail to adequately capture abrupt, localized anomalous dynamics. Such scenarios may lie beyond the current scope of our model and represent an important direction for future work, where more sophisticated constitutive relations and extended dynamical formulations could be incorporated.
> >
> > ---
> >
> > ### Q2. Practical guidance for grid size selection
> >
> > We appreciate this practical and insightful question. We propose the following physics-informed heuristic: set the grid cell size $\Delta x$ to be approximately equal to the typical displacement of a pedestrian within one time step. Formally, we recommend initializing:  $\Delta x \approx ||V\_{\text{mean}}||\_2 \cdot \Delta t,$
> > where  $||V\_{\text{mean}}||\_2 $ is the average speed magnitude in the training data and  $\Delta t$  is the inter-frame interval. This ensures sufficient cross-grid movements to generate meaningful flux signals for learning, while avoiding overly sparse or noisy graphs.

---

### Official Review · Reviewer_JPbf · 2025-11-02

**Soundness:** 4
**Presentation:** 3
**Contribution:** 4
**Rating:** 8
**Confidence:** 4

**Summary:**

This paper proposes STDDN (Spatio-Temporal Decoupled Differential Equation Network), a novel physics-guided deep learning framework for crowd simulation.
STDDN explicitly combines microscopic trajectory prediction with macroscopic density evolution by embedding the continuity equation from fluid dynamics into a Neural ODE structure.
The model separates local trajectory dynamics from global density fields, enabling physical consistency and stable long-term simulations.
Experiments on four real-world crowd datasets (GC, UCY, ETH, HOTEL) show that STDDN outperforms prior physics-guided baselines such as SPDiff and PCS in both accuracy and inference speed.

**Strengths:**

1.	Good motivation on coupling of micro- and macro-level dynamics.

The paper’s main contribution is conceptually sound. By using the continuity equation as a bridge between trajectory prediction and density evolution, STDDN unifies local motion modeling with global flow consistency.

2.	Physically meaningful ODE formulation.

The introduction of a Neural ODE to simulate density evolution is well justified. It provides continuous-time reasoning while enforcing conservation principles, addressing a key limitation of purely data-driven models that tend to accumulate errors over time.

3.	Strong empirical performance.

Across four datasets, STDDN shows consistent gains over all baselines, including both physics-based and deep learning methods. The improvements in both accuracy and latency demonstrate that the proposed framework is practically beneficial.

4.	Interpretability and efficiency.

The method retains interpretability through its physically grounded formulation while remaining computationally tractable, which is uncommon in physics-guided models.

**Weaknesses:**

1.	Limited experimental diversity.

Although the method is tested on multiple datasets, all belong to similar crowd domains. It would strengthen the generality claim to include different physical systems, such as vehicle or swarm simulation.

2.	Ablation breadth.

The ablation study is informative but it would be useful to show how performance changes under different ODE solvers or with alternative coupling strengths between density and trajectory modules.

3.	Minor missing citations for ODE-based trajectory forecasting.

The paper would benefit from acknowledging prior studies that have already explored ODE formulations for trajectory or crowd prediction, such as Social ODE: Multi-agent Trajectory Forecasting with Neural Ordinary Differential Equations (ECCV 2022) and Improving Transferability for Cross-Domain Trajectory Prediction via Neural Stochastic Differential Equation (AAAI 2024).
These works share conceptual overlap in embedding physical dynamics into continuous differential frameworks.

**Questions:**

Please see the weakness section

---

> ### Author Response · Authors · 2025-11-21
>
> Thank you very much for your insightful review and for your generous recognition of our work. Your constructive feedback has significantly helped us improve the completeness and rigor of the paper. Below, we address each of your concerns in detail.
>
> ---
>
> ### W1. Limited experimental diversity
>
> Thank you for raising this important point. Compared to open crowd scenarios, road traffic systems exhibit stronger structural regularity—their graph structures are typically sparser and more static—posing distinct modeling requirements for dynamic graph construction. Designing more efficient and scalable graph-generation strategies tailored to such physical systems is a promising direction for future research, and we have explicitly highlighted it as a key avenue for future work in the revised manuscript.
>
> The current study focuses on crowd datasets primarily due to the nature of our dynamic graph construction mechanism. Specifically, our method builds a time-varying graph based on spatial discretization into grids, and the size of the adjacency matrix is tightly coupled with the grid resolution. In large-scale scenarios such as vehicle traffic, maintaining sufficient spatial resolution would require significantly denser grids, leading to a dramatic increase in adjacency matrix dimensionality and consequently imposing prohibitive computational and memory costs. Therefore, this paper first validates the proposed “microscopic–macroscopic coupling” modeling paradigm in crowd scenarios to ensure the feasibility and practicality of our approach.
>
> ---
>
> ### W2. Ablation breadth
>
> We greatly appreciate this constructive suggestion. We fully agree that a more comprehensive ablation study would better clarify the role of each component. In response, we have added the following analyses:
>
> 1. Performance comparison under different ODE solvers (Euler, RK4, and Dopri5);
> 2. Alternative density–velocity coupling strategies, including:
>    - Removing the dynamic adjacency matrix and bias term (“w/o NE”)
>    - Replacing the learned weight matrix with an attention-based mechanism (“Trans”)
>
>
> | Method          | GC (MAE) | GC (OT) | UCY (MAE) | UCY (OT) |
> |-----------------|----------|---------|-----------|----------|
> | Dopri5          | 1.1315   | 1.7422  | 1.9654    | 5.2318   |
> | RK4             | 1.2311   | 1.8625  | 2.0381    | 5.4581   |
> | w/o NE          | 0.8921   | 1.3881  | 1.7917    | 3.7131   |
> | Trans           | 0.8901   | 1.3611  | 1.7833    | 3.7055   |
> | **Ours (Euler)**| **0.8875**| **1.3582**| **1.7747**| **3.6503**|
>
>
> Our results show that higher-order solvers (e.g., RK4, Dopri5), while theoretically more accurate, degrade performance in our setting. This is because our model fundamentally captures discrete frame-to-frame transitions on spatio-temporal grids, rather than a smooth continuous flow. Higher-order solvers introduce interpolated intermediate states that do not correspond to actual observations, increasing computational cost and risking overfitting. In contrast, the Euler method aligns naturally with our discrete autoregressive formulation, achieving the best trade-off between accuracy and efficiency.
>
> Furthermore, all alternative coupling strategies underperform our proposed physics-informed dynamic graph design, further validating its effectiveness. We believe these new ablations provide stronger evidence for the rationale behind our architectural choices. The results will be incorporated into the ablation study section of the revised manuscript. Thank you again for this valuable suggestion.
>
>
>
> ---
>
> ### W3. Minor missing citations for ODE-based trajectory forecasting
>
> We sincerely thank you for pointing us to these relevant works. In the revised manuscript, we will systematically acknowledge and cite these important contributions, including but not limited to [1]–[4]. While these methods also employ Neural ODEs to model continuous-time dynamics, our work is distinct in that we are the first to embed the macroscopic continuity equation as a differentiable, end-to-end computational module within a Neural ODE framework, specifically tailored for long-horizon crowd simulation that jointly models microscopic trajectories and macroscopic density evolution.
>
> **References:**
>
> 1. Liang, Y., Ouyang, K., Yan, H., et al. (2021). Modeling trajectories with neural ordinary differential equations. *IJCAI*, 1498–1504.
> 2. Wen, S., Wang, H., & Metaxas, D. (2022). Social ODE: Multi-agent trajectory forecasting with neural ordinary differential equations. *ECCV*.
> 3. Park, D., Jeong, J., & Yoon, K.-J. (2024). Improving transferability for cross-domain trajectory prediction via neural stochastic differential equation. *AAAI*.
> 4. Ke, K., et al. (2024). Social LODE: Human trajectory prediction with latent ODEs. *ICASSP*.
>
>
> ---
>
> Thank you once again for your thoughtful and helpful feedback, which has greatly strengthened our paper.

---

### Author Response · Authors · 2025-12-02
**Summary for Area Chair**

Dear Area Chair,

We fully understand the heavy workload you are facing during this busy decision-making period, and we sincerely appreciate the time and effort you have dedicated to maintaining a high-standard review process.

To assist you in your final evaluation, we provide the following concise summary of our paper’s contributions and the consensus reached during the review process:

- **Paper Overview and Consensus**
  This paper proposes **STDDN**, a physics-informed crowd simulation framework that embeds the continuity equation into a neural ordinary differential equation (Neural ODE) to jointly model microscopic trajectories and macroscopic density evolution. Reviewers generally acknowledged its physical plausibility and empirical advantages. During the rebuttal phase, we substantially strengthened the manuscript by adding extensive experiments—including new ablation studies, results on additional datasets, multiple physics-based consistency metrics (e.g., collision rate, density extreme analysis), training cost analysis, and theoretical clarifications—effectively addressing all technical concerns. Notably,  reviewer aHns **raised their score from 4 to 6 on 26 Nov, EST** ( **prior** to the OpenReview malfunction that occurred around 27 Nov, 10AM EST ) after our revised submission, indicating their endorsement.

- **Initial Scores**: 8, 6, 6, 4
---
- **Recognized Contributions**
  - A physically interpretable neural framework that couples micro- and macro-scale dynamics;
  - Four key modules that effectively integrate fluid conservation laws into a deep learning architecture;
  - State-of-the-art performance across four real-world crowd datasets, demonstrating clear advantages in both prediction accuracy and inference efficiency.

- **Score Updates & Key Discussion Points**
  - **Reviewer JPbf (8)**: Praised the method design and results, and suggested additional ablation studies and related work citations. We have incorporated these suggestions.
  - **Reviewer Fbqh (6)**: Recognized the novelty and clarity of the paper, and recommended including direct physics-based metrics and training cost analysis. We added comprehensive evaluations in the revised manuscript.
  - **Reviewer n74p (6)**: Appreciated the theoretical rigor and technical sophistication, and suggested discussing boundary conditions and expanding ablation studies. We have added discussion on open-boundary modeling and corresponding experiments.
  - **Reviewer aHns (4 → 6)**: Raised concerns about method novelty, implementation details, and comparisons with trajectory prediction SOTA methods. We **clarified that our work focuses on long-term crowd simulation (not short-term trajectory prediction)**, provided a detailed explanation of how the continuity equation is implemented within the Neural ODE framework, corrected figure and formula presentation issues, and reported ADE/FDE results on ZARA1 and ZARA2. Following these revisions, the reviewer updated their score to 6.
---
We believe that our thorough rebuttal and supplementary experiments—including results on additional datasets, physics-based and trajectory evaluation metrics, and ablation studies—have fully addressed the reviewers’ initial concerns. We hope this summary is helpful for your decision.

Thank you again for your invaluable contributions to the community.

With sincere gratitude,  The Authors

---

### Meta-Review · Area_Chair_kq5Q · 2026-01-06

**Summary:**

This paper proposes STDDN, a physics-guided crowd simulation framework that couples microscopic trajectory rollout with macroscopic density evolution by embedding the continuity equation into a Neural ODE.

Reviewers found the approach conceptually sound and physically interpretable, with well-designed differentiable density/flux modules, strong long-horizon results on four real-world datasets, and clear inference-efficiency benefits. In rebuttal, the authors effectively addressed key concerns by adding direct physics metrics (e.g., collision/density analyses), expanded ablations (solver/coupling variants), reporting training cost, and clarifying how the continuity constraint is discretized and implemented in the ODE; they also improved per-dataset reporting and related-work positioning.

Overall, the technical case is solid and substantially strengthened after rebuttal. Recommendation: accept.

**Reviewer Concerns:**

- Generalization beyond crowd scenarios (e.g., traffic/swarms) remains future work.
- Open-world dynamics (explicit source/sink terms for entries/exits) are discussed but not fully modeled.
- The necessity of a continuous-time ODE vs. an equivalent discrete formulation is clarified but not conclusively justified as essential.

**Reviewer Scores:**

N.A.

---

### Decision · Program_Chairs · 2026-01-26

Accept (Poster)